# Towards understanding the optimization landscape of GRPO and its variants

## Abstract

GRPO has achieved impressive success in the landscape of reasoning models. However, the motivation behind its origins along with the reasons for its effectiveness remain elusive. In this work, we fill some of the gaps and demonstrate that in on-policy setting, GRPO's optimization can be viewed as a weighted combination of maximization of likelihood for correct rollouts and minimization for the incorrect ones. This finding gives a different perspective about the optimization landscape of GRPO. Motivated by this, we analyze the positive and negative part of GRPO's objective function independently, and find that their global minima correspond to undesired solutions. While optimization of the positive term leads to entropy minimization and length collapse, optimizing for the negative term leads to entropy maximization and length explosion. Using this lens, we show the presence of instability in on-policy training of some recent algorithmic advances trying to simplify GRPO's objective. However, despite the presence of bad global minima in GRPO's objective function, it doesn't converge to either of them. We identify design choices in GRPO's advantages that aid convergence of GRPO to good minima. We also demonstrate the effectiveness of using clipping in stabilizing the optimization process, thereby preventing training instabilities even when training only for minimizing the likelihood of incorrect rollouts.

## 1 Introduction

Reinforcement learning with verifiable rewards (RLVR) has shown impressive improvements in the reasoning abilities of Large Language Models (LLMs) on tasks like maths, coding, etc. DeepSeek-AI (2025); Jaech et al. (2024); Team et al. (2025). These improvements are a result of advancements in the capabilities of the base models, along with development of improved RL algorithms like GRPO (Shao et al., 2024) and PPO (Schulman et al., 2017b). It is interesting that merely using rewards at the end of model's prediction (output verifiable rewards) is sufficient to observe non-trivial improvements in reasoning abilities of LLMs. This observation has led to a large adoption of outcome reward models (ORMs) in the community resulting in increased popularity of RL finetuning methods like GRPO. As a result several modified versions of GRPO have been proposed recently, aiming to improve the training efficiency and generalization of RL-finetuned models.

Amongst these alternatives, a few of them have tried simplifying GRPO's objective function. For instance, (Xiong et al., 2025) demonstrated that very simple algorithm RAFT (and its enhancement RAFT++) that trains only on the correct rollouts give performance comparable with GRPO. On the other hand, Zhu et al. (2025) demonstrated that minimizing the likelihood only on the incorrect rollouts performs comparably to GRPO while improving model's output diversity. Additionally, Samineni et al. (2025) demonstrated that a simple combination of the positive and negative losses as described above, also leads to performance similar to GRPO. A few of the recently proposed alternatives have also tried modifying the clipping mechanisms in GRPO (MiniMax, 2025; Ahmadian et al., 2024). Similarly, Zheng et al. (2025); Zhao et al. (2025c) modify GRPO's objective function by considering importance sampling at sequence level. All these methods amongst several others (Chen et al., 2025a; Lanchantin et al., 2025) aim to simplify GRPO's objective function and they have successfully demonstrated their effectiveness on certain datasets and base models. These results point to the lack of motivation about different design choices in GRPO's objective function.

Another line of works modifies the definition of advantages and rewards in GRPO: Chen et al. (2025b); Mahdavi et al. (2025); Zhou et al. (2025); Liu et al. (2025); Zhao et al. (2025b); Kang et al. (2025);

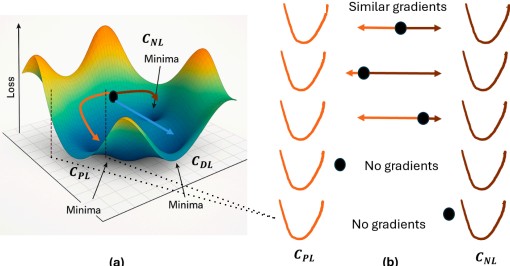

Figure 1: **Illustrative diagram demonstrating the loss landscape of RLVR methods:** (a) The loss landscape consists of different critical solutions $C_{PL}$, $C_{\mathrm{NL}}$, and $C_{DL}$, where $C_{PL}$ represents the minimum entropy solution, $C_{\mathrm{NL}}$ represents the maximum entropy solution, and $C_{DL}$ leads to improved performance of the model. (b) GRPO stabilizes the model against converging to bad critical solutions $C_{PL}$ and $C_{\mathrm{NL}}$, by controlling the magnitude of the gradient as shown by the length of the arrows, and the illustration of the model in its function space is shown by the dark circle.

Arora & Zanette (2025); He et al. (2025); Prabhudesai et al. (2025); Xiao et al. (2025); Hao et al. (2025); Fan et al. (2025); Yu et al. (2025); Shafayat et al. (2025); Arnal et al. (2025). In particular, Prabhudesai et al. (2025) show that using GRPO in unsupervised RL setting by minimizing model's entropy could lead to improved performance. On the other hand, Wang et al. (2025b) demonstrated that maximization of entropy could also lead to improved performance. These contrasting results have created confusion within the community, highlighting a need to understand the optimization landscape of GRPO and related simplified objective functions.

In this work, we primarily focus on understanding the role of algorithmic advancements in enabling improvements in reasoning abilities of LLMs. For this, we first try to understand the motivation behind each of the design choices used in GRPO's objective function, where we find that several of them lack adequate motivation. We first highlight the lack of motivation behind using clipping in GRPO's objective function by tracing back the origins of clipping proposed in PPO (Schulman et al., 2017b), which had its motivation grounded in policy improvement guarantee shown in TRPO (Schulman et al., 2017a). Thus we begin by analyzing GRPO's objective function in on-policy setting, and adding different design choices to reconstruct its actual objective function. In the on-policy setting, we demonstrate how several approximations inherently made by GRPO in comparison to PPO, simplify its learning process to a reweighted version of maximization and minimization of likelihood on the correct and incorrect rollouts respectively. Using this perspective, we analyze the loss on samples with correct and incorrect answers separately, and characterize the properties of their corresponding critical solutions. Let us term the critical solutions corresponding to maximizing and minimizing the likelihood for positive (correct) and negative (incorrect) samples respectively as $C_{PL}$ and $C_{\mathrm{NL}}$ (as shown in Fig. 1 (a)). Empirically, we observe that converging to either of these minimas results in degraded performance, where training on the correct rollouts leads to collapse of entropy and length of model's outputs, and training on the incorrect rollouts leads to explosion of entropy and length of model's outputs. As GRPO's gradients can be considered a weighted mixture of the gradients of these losses, we aim to understand the reweighting mechanism which prevents GRPO from converging to either of the two bad solutions. As shown in Fig. 1 (b), we find that the advantages in GRPO help in reducing the norm of gradients when the model comes closer to either of the minimas in its function space, while increasing the norm of the gradient in the direction of farther away critical point.

Specifically, we find that the on-policy versions of the algorithms in Zhu et al. (2025); Xiong et al. (2025) are prone to instability and collapses, as explained by convergence to $C_{PL}$ and $C_{\mathrm{CL}}$ above (these works focus their evaluations in off-policy setting without discussing the on-policy vs. off-policy distinction). However, utilizing clipping, makes the training stable for these methods, even though there is no policy improvement guarantee as in case of TRPO. Overall this indicates the critical role played by clipping in enabling off-policy learning to become more stable than on-policy, even in cases where the algorithms do not maximize a strict lower bound on the value function of the states in MDP, and therefore do not enjoy any policy improvement guarantee as in case of TRPO.

Designing more stable and robust RLVR methods is important from a practical viewpoint, and in an attempt to improve training stability of existing methods in on-policy setting, we find that utilizing

normalization at token level could be helpful. We further explain the reasons behind this, thereby providing a new perspective on using token level normalization instead of sequence level, which has recently gained traction in the community as well Yu et al. (2025); Liu et al. (2025); Yue et al. (2025b). To summarize, our key contributions are:

- We provide a new view on GRPO as a reweighted version of maximization and minimization of likelihood for correct and incorrect samples.
- We characterize the properties of the critical solutions of the two minimas corresponding to maximization and minimization of likelihood for correct and incorrect samples, respectively.
- We show that clipping and utilizing token level normalization help off-policy training become more stable against collapsing as opposed to on-policy settings for different variations of GRPO.
- We demonstrate the key role played by the advantages used in GRPO in stabilizing the training.

We defer the detailed discussion of related work and background to Appendix A and B respectively.

## 2 UNDERSTANDING THE ORIGINS OF GRPO

We consider an episodic MDP given by the tuple $(S, \mathtt{A}, P, r, \gamma)$, where $S$ is a set of states and $\mathtt{A}_{s_t}$ is a set of actions allowed for a given state $s_t$. We assume same action set for all states gives us $\mathtt{A}_{s_0} = \mathtt{A}_{s_1} = ... = \mathtt{A}_{s_T} = \mathtt{A}$. The policy is parametrized by $\theta$ and defined as $\pi_\theta : R^d \rightarrow [0, 1]^v$, where $v$ is the cardinality of $\mathtt{A}$ and $d$ is the dimension of the input. Denote by $P : S \times \mathtt{A} \times S \rightarrow R$ the transition probability matrix, by $r_t$ the reward given by the environment at time stamp $t$. Let the process be episodic and always start from a state $s_1$. We can now define the advantage function ($A_\pi(s_t, a_t)$) for our policy $\pi_\theta$: $A_\pi(s_t, a_t) = q_\pi(s_t, a_t) - v_\pi(s_t) = q_\pi(s_t, a_t) - \sum_{a_i} \pi_\theta(a|s_t)q_\pi(s_t, a_i)$, where $q_\pi(s_t, a_t)$ represents the Q-value function calculated at state $s_t$ and for action $a_t$.

While the above formulation can be used for general MDPs, in the case of language models we get a special MDP: The input prompt $q$ denotes the starting state given by $s_1$. An action refers to prediction of the next token and a state is obtained by appending the predicted token (i.e. $s_t = (q, a_1, ..., a_{t-1})$). Given the current state, the next state is deterministically determined by the action. A response is a set of actions given by $a = (a_1, a_2, ..., a_T)$, where the prediction of end of sequence (EOS) token determines the end of episode. We use verifiable rewards given at the end of the episode, where correct prediction results in a reward score of one, and an incorrect prediction results in zero reward. We will now derive GRPO using PPO as the base method, where PPO's objective function is:

$$J(\theta, q) = \mathop{\mathbb{E}}_{a \sim \pi_{\theta_{\text{old}}}(a|q)} \frac{1}{|a|} \sum_{t=1}^{|a|} \min[\frac{\pi_\theta(a_t|s_t)}{\pi_{\theta_{\text{old}}}(a_t|s_t)} A_\pi(s_t, a_t), \text{clip}(\frac{\pi_\theta(a_t|s_t)}{\pi_{\theta_{\text{old}}}(a_t|s_t)}, 1-\epsilon, 1+\epsilon) A_\pi(s_t, a_t)] \quad (1)$$

where $\pi_\theta$ and $\pi_{\theta_{\text{old}}}$ represent the current and the old policy utilized for sampling the rollouts. PPO utilizes generalized advantages ($A_\pi(s_t, a_t)$) which are motivated from TD learning (Sutton, 1988). Training of the policy can be done in either off-policy or on-policy setting. In the on-policy setting the same model is used for training and generating the rollouts (i.e., $\pi_{\theta_{old}} = \pi_\theta$), whereas in the off-policy setting an older version of our current policy is utilized for generating the rollouts.

GRPO builds on Eq. 1 and modifies the calculation of advantages. It approximates a) the computation of Q-value functions ($Q(s_t, a_t)$) by a single rollout, which results in an unbiased estimator with high variance. b) It also assumes that the Q-value and the value functions ($V(s_t)$) are the same for all the states in a rollout, where the value function is calculated by generating rollouts only at the starting state ($s_1$). This gives

$$A_\pi(q, a) = A_\pi(s_1, a_1) \stackrel{\text{assumes}}{=} A_\pi(s_T, a_T) = Q(s_1, a_1) - V(s_1) \stackrel{\text{assumes}}{=} r(a) - V(s_1) = r(a) - V(q) \quad (2)$$

GRPO also divides $A_\pi(s_{t+1}, a_{t+1})$ by the standard deviation of the rewards for the sampled trajectories which lacks desired motivation. Using the highlighted assumptions, we can now rewrite Eq. 1 in the following way:

$$J(\theta, q) = \mathop{\mathbb{E}}_{a \sim \pi_{\theta_{\text{old}}}(a|q)} \frac{1}{|a|} \sum_{t=1}^{|a|} \min[\frac{\pi_\theta(a_t|s_t)}{\pi_{\theta_{\text{old}}}(a_t|s_t)} \tilde{A}_\pi(q, a), \text{clip}(\frac{\pi_\theta(a_t|s_t)}{\pi_{\theta_{\text{old}}}(a_t|s_t)}, 1-\epsilon, 1+\epsilon) \tilde{A}_\pi(q, a)] \quad (3)$$

where $\tilde{A}_\pi(q, a) = \frac{A_\pi(q,a)}{\mathbb{E}_{\tilde{a} \sim \pi_{\theta_{\text{old}}}(\tilde{a}|q)}(r(\tilde{a}) - V(q))^2}$ Note that we do not incorporate the KL terms here as recent works: Zhou et al. (2025); Hu et al. (2025) have shown that using GRPO without the

KL divergence results in faster convergence and improved performance. In eq. 3, GRPO further approximates the value function using a group of a few rollouts.

## 2.1 ON-POLICY GRPO AND ITERATED MLE

GRPO was motivated by PPO which in turn was obtained by modification of TRPO (Schulman et al., 2017a) whose history goes further back. However, the conceptual underpinnings of GRPO become somewhat obscured because of this long chain. We elaborate this further. TRPO enjoys a policy improvement guarantee in each iteration, by formalizing a constrained maximization of a strict lower bound on the value function, where the constraint ensures that the policy remains in close proximity of the old policy. PPO further proposes clipping as a heuristic to realize the constraint in practice. This led PPO enhance training stability of deep RL methods including the on-policy ones. However, due to the approximations detailed in Sec. 2, GRPO and its simplified versions no longer maximize a strict lower bound on the value function of the states in MDP in every iteration. This means that in an iteration some states might increase their value function while others might decrease it. As a result, there is no policy improvement guarantee, which fades away the motivation to utilize clipping, as used in PPO. This motivates us to unwrap different design choices in GRPO, and analyze their role in stabilizing training. Thus we start by investigating GRPO in the on-policy setting. First, we find that on-policy version of GRPO can be considered as an arguably simple and natural algorithm we call *iterated MLE*. Consider the following simple iterated maximum likelihood optimization algorithm: in each iteration, sample a prompt $q$, and then sample the rollouts $a$ for this prompt. We form an expression for likelihood by taking the positive sign for $a$ having the correct outcome and the negative sign for $a$ having the incorrect outcome (specified by $A_\pi^{\text{sign}}(q, a)$ taking values $+1$ and $-1$, resp.).

$$L^{\text{IMLE}}(\theta, q) = \mathop{\mathbb{E}}_{a \sim \pi_\theta(a|q)} \sum_{t=1}^{|a|} A_\pi^{\text{sign}}(q, a) \log[\pi_\theta(a_t|s_t)]. \tag{4}$$

$$\theta_{i+1} = \theta_i + \alpha \mathop{\mathbb{E}}_{a \sim \pi_\theta(a|q)} \sum_{t=1}^{|a|} A_\pi^{\text{sign}}(q, a) \nabla \log[\pi_\theta(a_t|s_t)] = \theta_i + \alpha \mathop{\mathbb{E}}_{a \sim \pi_\theta(a|q)} \sum_{t=1}^{|a|} A_\pi^{\text{sign}}(q, a) \frac{\nabla \pi_\theta(a_t|s_t)}{\pi_\theta(a_t|s_t)} \tag{5}$$

In the case of GRPO we can write the gradients for Eq. 3 in the on-policy setting ($\pi_\theta = \pi_{\theta_{\text{old}}}$) as:

$$\theta_{i+1} = \theta_i + \alpha \mathop{\mathbb{E}}_{a \sim \pi_\theta(a|q)} \frac{\alpha}{|a|} A_\pi(q, a) \sum_{t=1}^{|a|} \frac{\nabla \pi_\theta(a_t|s_t)}{\pi_\theta(a_t|s_t)}. \tag{6}$$

By comparing Eq. 5 with Eq. 6, it is clear that we can consider GRPO as doing reweighted version of iterated MLE, where the reweighing of gradients is done at sample level with weight given by $\frac{|A_\pi(q,a)|}{|a|}$. Importantly, note that the sign of $A_\pi(q, a)$ agrees with $A_\pi^{\text{sign}}(q, a)$.

Note that when compared with classic on-policy methods like Reinforce, the key difference here is that in case of GRPO, the reweighting mechanism acts at sample level, where as in traditional on-policy objectives reweighting (using advantages) is done at token level. This distinction precisely occurs due to approximations inherently adopted by GRPO as discussed in Sec. 2. Further, this distinction inhibits GRPO from maximizing a strict lower bound of the true value functions of the states in the MDP.

In summary, we have:

> **Takeaway 1**
>
> In the on-policy setting, GRPO behaves like reweighted iterated MLE.

With this new perspective, we will now try to uncover the role the reweighting mechanism and clipping used in GRPO's objective function. But first we define our experimental setup below.

## 3 EXPERIMENTAL SETUP

To make our findings robust across different settings, we apply various RLVR finetuning methods (GRPO, $L_{\text{PL}}$, $L_{\text{NL}}$, $L_{CL}$, PPO) on several training datasets: SimpleRL (Zeng et al., 2025), Countdown (Pan et al., 2025), Numina-Math (LI et al., 2024), and Numina-Math Hard which we specially

crafted ourselves by filtering the prompts in Numina-Math that fail at Pass@2 when evaluated using Qwen2.5-7B (Qwen et al., 2025). More datasets details are in App. L. We also use multiple models: Qwen2.5-7B, Qwen2.5-7B-Instruct, and Llama3.1-8B-Instruct (Grattafiori et al., 2024). In most of the paper we plot the evolution of accuracy on train set of corresponding runs. The reader will notice that some of the plots are missing; the corresponding experiments had not been completed due to limited compute. Of course we are concerned about the test accuracy and we show in Table 1 that high train accuracy correlates with high test accuracy even across datasets different from the training dataset. For test set we use Math 500 (Hendrycks et al., 2021) , GSM8K (Cobbe et al., 2021), Minerva-Math (Lewkowycz et al., 2022), College-Math (col, 2024), OlympiadBench (He et al., 2024), and Gaokao-2023 (Zhang et al., 2024). Many of our plots involve a single model trained on multiple datasets (varying datasets), the base model in these plots is Qwen2.5-7B. Similarly, we have plots for varying base models for a fixed dataset SimpleRL. Ideally, one would try all combinations of datasets and models; this however is computationally infeasible. Experiments are run only once because of a large number of experiments. We use 8 rollouts per prompt, batch size of 128 prompts, training batch size of $128 \times 8$ samples for on-policy training, and $32 \times 8$ samples for off-policy training. For most other hyperparameters we use the default settings from VeRL (Sheng et al., 2024).

## 4 UNWRAPPING GRPO

In this section, we will perform several ablations on the design choices of GRPO to highlight the key ingredients behind GRPO's success. The ablations include training only on the correct rollouts or the incorrect ones, the use of advantages, on-policy and off-policy training with clipping. Since GRPO can be considered as a reweighted version of likelihood maximization and minimization on rollouts with correct and incorrect outputs respectively, we start by writing the empirical expectation of GRPO's objective function in Eq. 3 decomposed according to whether the response $a$ to $q$ has the correct outcome or not.

$$J(\theta, q) = \frac{1}{|\mathcal{A}^+| + |\mathcal{A}^-|} \left( \sum_{a \in \mathcal{A}^+} \frac{1}{|a|} \sum_{t=1}^{|a|} \min\left[ \frac{\pi_\theta(a_t|s_t)}{\pi_{\theta_{\text{old}}}(a_t|s_t)} A_\pi(q, a), \text{clip}\left(\frac{\pi_\theta(a_t|s_t)}{\pi_{\theta_{\text{old}}}(a_t|s_t)}, 1 - \epsilon, 1 + \epsilon\right) \right. \right.$$

$$\left. \left. A_\pi(q, a) \right] + \sum_{a \in \mathcal{A}^-} \frac{1}{|a|} \sum_{t=1}^{|a|} \min\left[ \frac{\pi_\theta(a_t|s_t)}{\pi_{\theta_{\text{old}}}(a_t|s_t)} A_\pi(q, a), \text{clip}\left(\frac{\pi_\theta(a_t|s_t)}{\pi_{\theta_{\text{old}}}(a_t|s_t)}, 1 - \epsilon, 1 + \epsilon\right) A_\pi(q, a) \right] \right) \quad (7)$$

where $\mathcal{A}^+$ corresponds to the set of correct rollouts, and $\mathcal{A}^-$ refers to the set of incorrect rollouts. Thus, $\mathcal{A}^+ \cup \mathcal{A}^- = \mathcal{A}$, where all the rollouts are sampled from $\pi_{\theta_{\text{old}}}$. Note that $a \in \mathcal{A}^+$ implies $A_\pi(q, a) \geq 0$, and $a \in \mathcal{A}^-$ implies $A_\pi(q, a) \leq 0$. To ablate the use of advantages in GRPO, we replace the advantages in equation 7 with their signs. (This gives a version of GRPO that's similar to iterated MLE but with clipping.) Let us name the objective corresponding to the set $\mathcal{A}^+$ as Positive Likelihood ($-L_{\text{PL}}$) and the objective corresponding to the set $\mathcal{A}^-$ as Negative Likelihood ($-L_{\text{NL}}$). In the on-policy setting, where $\pi_\theta = \pi_{\theta_{old}}$, these objectives simplify further as follows:

$$L_{\text{PL}}(\theta, q) = \frac{-1}{|\mathcal{A}^+|} \sum_{a \in \mathcal{A}^+} \frac{1}{|a|} \sum_{t=1}^{|a|} \frac{\pi_\theta(a_t|s_t)}{\pi_{\theta_{\text{old}}}(a_t|s_t)}; \ \nabla_\theta L_{\text{PL}} = \frac{-1}{|\mathcal{A}^+|} \sum_{a \in \mathcal{A}^+} \frac{1}{|a|} \sum_{t=1}^{|a|} \frac{\nabla_\theta \pi_\theta(a_t|s_t)}{\pi_\theta(a_t|s_t)} = 0 \quad (8)$$

$$L_{\text{NL}}(\theta, q) = \frac{1}{|\mathcal{A}^-|} \sum_{a \in \mathcal{A}^-} \frac{1}{|a|} \sum_{t=1}^{|a|} \frac{\pi_\theta(a_t|s_t)}{\pi_{\theta_{\text{old}}}(a_t|s_t)}; \ \nabla_\theta L_{\text{NL}} = \frac{1}{|\mathcal{A}^-|} \sum_{a \in \mathcal{A}^-} \frac{1}{|a|} \sum_{t=1}^{|a|} \frac{\nabla_\theta \pi_\theta(a_t|s_t)}{\pi_\theta(a_t|s_t)} = 0 \quad (9)$$

We note that the objective functions above have been analyzed in prior works Zhu et al. (2025); Xiong et al. (2025), where the authors indeed find them to be stable. This further motivates us to analyze these objectives in detail. We characterize the minimizers of the two objective functions. We first do this in an idealized setup where the response $a$ consists of only one token and moreover we use the expectation instead of the empirical mean used in the expressions for $L_{\text{PL}}$ and $L_{\text{NL}}$ above. We suggest that the minimization of $L_{\text{PL}}$ and $L_{\text{NL}}$ leads to the problem of minimizing and maximizing the entropy of the distribution of $a$, resp. Note that in Eq. 4, by decomposing the expectation into two

parts according to the correctness of the response $a$, we can express the objective as the difference of the entropies (denoted by $H$) on the two distributions of the two parts:

$$L^{\text{IMLE}}(\theta, q) = \Pr[A_\pi^{\text{sign}}(q, a) = 1]H[\pi_\theta(a|q, A_\pi^{\text{sign}}(q, a) = 1] -$$
$$\Pr[A_\pi^{\text{sign}}(q, a) = -1]H[\pi_\theta(a|q, A_\pi^{\text{sign}}(q, a) = -1] \quad (10)$$

Note also from Eqs. 8 and 9 that the gradients of $L_{\text{PL}}$ and $L_{\text{NL}}$ have the same form as that of $L^{\text{IMLE}}$ in Eq. 5. Now it's well known that the entropy of a probability distribution on a finite set is maximized for the uniform distribution and is minimized when the distribution is concentrated on a single point. From the above facts we infer that for the idealized case, $L_{\text{PL}}$ is minimized by a distribution concentrated on a single token, and $L_{\text{NL}}$ is minimized by the uniform distribution.

The experimental setup we study differs from the idealized setup above in the following ways: the number of tokens in the response is not limited to 1, and moreover only the empirical mean is used in the loss computation. Next, we hypothesize a generalization of the above idealized solution. Assuming that the cardinality of sets $\mathcal{A}^-$ and $\mathcal{A}^+$ is large enough, the global minima corresponding to Eq. 8 and Eq. 9 are given by:

$$S_{\text{PL}} : \pi_\theta(a_t|s_t) = 1 \ \forall a \in \mathcal{A}^+ \text{ and } t \le |a|; \quad S_{\text{NL}} : \pi_\theta(a_t|s_t) = \frac{1}{|V|} \forall a \in \mathcal{A}^- \text{ and } t \le |a|, \quad (11)$$

where $|V|$ represents the size of the vocabulary. Some further justifications are presented in App. K. When we also drop the assumption of working with the full expectation and work with the empirical mean, we get the emprically realizable versions of the above solutions which we call $C_{\text{PL}}$ and $C_{\text{NL}}$. We will see below that empirically these have properties similar to their idealized counterparts:

$C_{\text{PL}}$: **Entropy:** Leads to minimum entropy of the model. **Output Length:** As the solution doesn't depend on the output length, the model can learn a shortcut by predicting the answer directly without the reasoning traces. **Stability:** Highly stable as the same mean has low variance as compared to population even for sample sizes of one. Refer to App. K for more discussion.

$C_{\text{NL}}$: **Entropy:** Leads to maximum entropy of the model. **Output Length:** Leads to longer sequence length as the probability for predicting the end of sequence token also becomes close to $\frac{1}{|V|}$. **Stability:** Highly unstable as the sample mean has large variance with respect to population mean. This is because we are sampling from a uniform distribution. Refer to App. K for more discussion.

Both the solutions $C_{\text{PL}}$ and $C_{\text{NL}}$ results in undesired behaviors. We demonstrate the same below.

## 4.1 ON-POLICY LEARNING

Training is performed to minimize $L_{\text{PL}}$ and $L_{\text{NL}}$, and the evolution of accuracy on train set is shown in Figs. 2 (a, b) and 2 (c, d) respectively. We validate our results across different models as well as datasets. As observed in these plots the model indeed learns $C_{\text{PL}}$ and $C_{\text{NL}}$ when trained on their corresponding objective functions. In addition to the sudden drop in model's performance, we also observe significant decease and increase in length of model's entropy when optimizing $L_{\text{PL}}$ and $L_{\text{NL}}$ respectively (See Fig. 12). The drop in entropy is accompanied by a drop in length of model's outputs, whereas increase in model's entropy leads to an increase in the length of model's outputs (See Fig. 11). Note that these observations are in accordance with Setlur et al. (2025), He et al. (2025). However, in this work, we move a step further, and investigate the stability and characterization of the solutions learned when collapse occurs, which we discuss next.

It is evident from comparison of PL and NL on QwenIT 7B in Fig. 2 (c, d) that the stability of the minima corresponding to $L_{\text{PL}}$ and $L_{\text{NL}}$ differ significantly from each other. $C_{\text{PL}}$ is very stable and as a result the model doesn't escape it once learned. Thus, we do not observe sudden jumps after the collapse, and the traces remain similar in nature as shown in Fig. 4(a). Here, the model indeed learns the shortcut by outputting the final answer directly without any chain of thoughts. On the other hand, the critical solution corresponding to $C_{\text{NL}}$ is not stable, and the model tends to escape it. This is evident from the traces in Fig. 4, 16, which change their nature very quickly and this is also followed by sudden changes in model's performance. The model sometimes quickly regains its performance (as shown in Fig. 16), while sometimes it is not able to retain its performance, but instead learns to output almost all constant tokens (as evident from traces shown in Fig. 16 (3), Fig. 4(b) (3)).

The observed instability is a result of using a few rollouts, which leads to a high variance of sample mean as compared to population mean of Eq. 9. Note that this is not the case with $C_{\text{PL}}$ as it will have

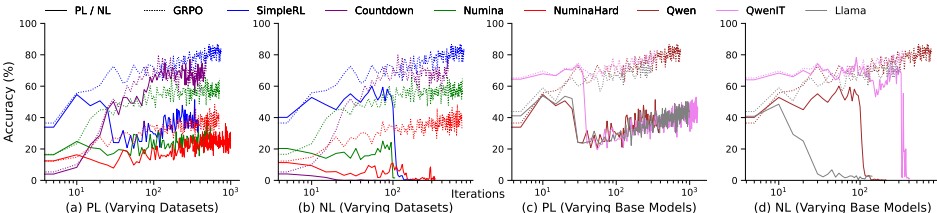

Figure 2: **On-Policy Experiments:** Comparison between the evolution of training accuracy for PL and GRPO (a, c) and NL and GRPO (b, d). Utilizing PL (a, c) and NL (b, d) losses leads to collapses across different datasets (a, b) and models (c, d)

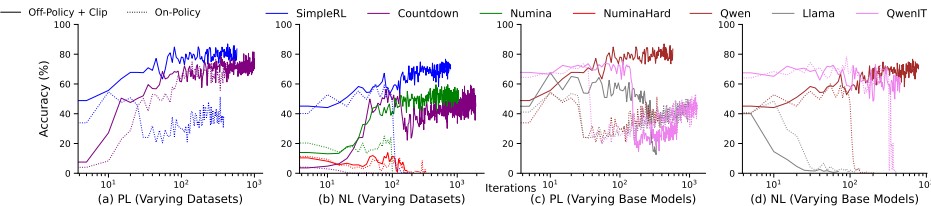

Figure 3: **Off-Policy with Clipping Experiments:** Comparison between the evolution of training accuracy for on- and off-policy PL (a, c) and on- and off-policy NL (b, d). Utilizing clipping with off-policy either delays or prevents the collapses observed in on-policy training.

low variance with respect to population mean even for single sample sizes. As a result, $C_{\mathrm{NL}}$ won't have zero gradient, despite having a low value of loss. In fact, the norm of gradients here becomes extremely large (as shown in Fig. 13 (b,d)), which makes the model converge into the solution space of functions outputting random tokens with high likelihood. This results in small gradient norms (even for small sample sizes). We summarize our findings in this section below:

---

**Takeaway 2**

- $L_{\mathrm{PL}}$ minimization leads to collapse: the model converges to a bad critical solution characterized by sudden loss of entropy and length of model's outputs.

- $L_{\mathrm{NL}}$ minimization also leads to collapse: explosion of entropy followed by a sudden increase in length of model's outputs. However this solution is not stable.

---

## 4.2 OFF-POLICY LEARNING

Tracing back to our motivation for investigating different design choices of GRPO, here we try to unwrap the role of clipping. As highlighted in Sec. 2.1, clipping is known to induce stability in PPO, but GRPO and its simplified versions do not maximize a strict lower bound on the value function of the states in MDP. Thus, there is no guarantee that utilizing clipping in off-policy setting would lead to policy improvement. This leads to unclear motivation to use clipping with GRPO and its simplified variants. To understand this in detail, we perform experiments same as Sec. 4.1 in the off-policy setting, with and without the use of importance sampling and clipping in Fig. 3 and Fig. 9 respectively.

Generally, off policy learning is known to be more unstable than on-policy in scenaiors where we utilize function approximations and bootstrapping (deadly triad (Sutton & Barto, 2018)), which we indeed observe in Fig.9, where off-policy setting makes the training even more unstable and leads to faster collapses. Next, we analyze the effect of incorporating importance sampling with clipping in Fig. 3. We find that, the collapses disappear and the training becomes stable. We also find that the stability observed on incorporating clipping is significantly better than the on-policy setting. This highlights that use of clipping remains crucial to induce stability in simplified objectives of GRPO: (Zhu et al., 2025; Xiong et al., 2025). Next, we dive into investigating if clipping indeed helps in improving stability of GRPO, which we expect to be helpful based on these results.

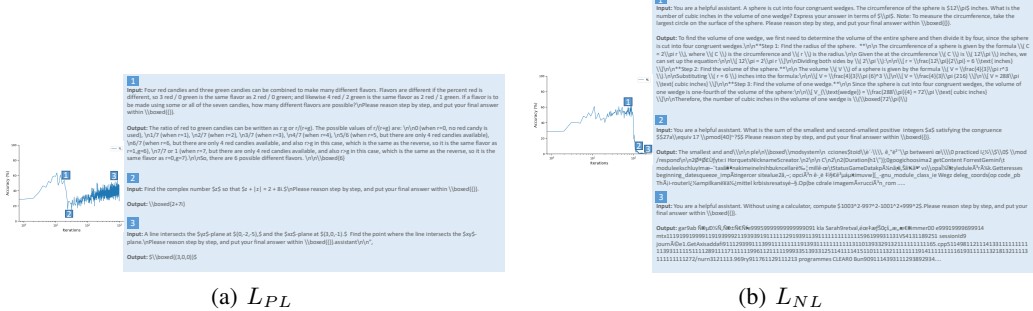

(a) $L_{PL}$             (b) $L_{NL}$

Figure 4: **Analysis of rollouts using** $L_{PL}$ **and** $L_{NL}$ objectives for Qwen2.5-7B in the on-policy setting on SimpleRL. a) Using $L_{PL}$ collapses the entropy and also the output's length. This makes the model output the final answer without any reasoning steps. b) In contrast, on using $L_{NL}$, the model tends to output random tokens first (2), which leads to enhanced length of the output. However, due to instability associated with this solution, the model ends up outputting repetitive tokens (3), thereby changing the nature of the rollouts.

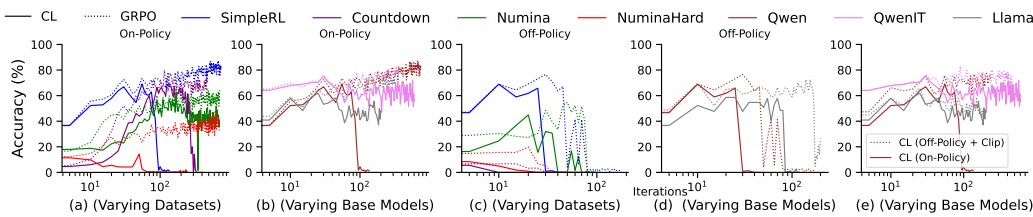

Figure 5: **Understanding the role of advantages in GRPO:** GRPO is robust against collapses in on-policy setting when compared with CL (a, b). In case of off-policy setting GRPO indeed collapses but the collapse is delayed as compared to CL (c, d). However, when using clipping, we observe CL to also become stable (e). This highlights the enhanced stability achieved due to clipping.

We find that using GRPO in on-policy setting with verifiable rewards remains significantly stable and on-par with GRPO in off-policy setting with clipping, (See Fig. 2). This highlights that clipping is not the key ingredient which helps in stabilizing GRPO, as opposed to other algorithms like PPO Schulman et al. (2017b), and other simplified versions of GRPO: Zhu et al. (2025); Xiong et al. (2025).

This is somewhat surprising, as even though the optimization of GRPO can be considered as reweighted combination of the gradient descent corresponding to $L_{PL}$ and $L_{NL}$ with bad critical points, it remains stable. Therefore, understanding the reweighting mechanism of GRPO becomes imperative.

### 4.3 UNDERSTANDING THE REWEIGHTING MECHANISM OF GRPO

Now we will analyze the effect of combining the two losses $L_{PL}$ and $L_{NL}$ in an effort to move closer to GRPO's objective function. Let us define the combined loss as $L_{CL} = \frac{|\mathcal{A}^+|L_{PL}+|\mathcal{A}^-|L_{NL}}{|\mathcal{A}|}$. Using the characteristics defined for the two critical solutions in Sec. 4, we show that the two critical solutions $C_{PL}$ and $C_{NL}$ discussed above, are critical solutions of $L_{CL}$ as well (See App. K for more details).. To analyze if the model converges to either of them, we train the policy to optimize $L_{CL}$. Comparing Fig. 5 (a, b) with Fig. 2, it is clear that in on-policy setting, optimizing $L_{CL}$ leads to enhanced stability as compared to optimizing $L_{PL}$ or $L_{NL}$ alone. However, using $L_{CL}$ still ends up collapsing (in most cases) if the training is continued for longer time. Similar results hold for off-policy setting without using clipping and importance sampling (See Fig. 5 (c, d)). But on using the reweighting mechanism in GRPO, the training becomes stable and almost never collapses. Similarly, in off-policy setting when not utilizing clipping and importance sampling, the collapse is delayed (See Fig. 5 (c, d)). This highlights the implicit stabilizing mechanism induced by the reweighting given by advantages in GRPO's optimization. We try to uncover this mechanism below.

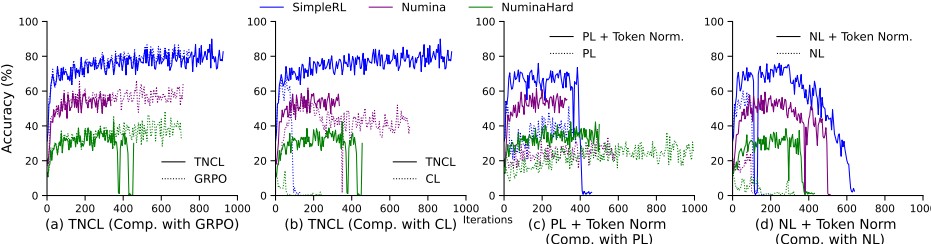

Figure 6: **Token normalization improves training stability:** (a) Using token normalization with CL enhances stability. (b) The training performance on SimpleRL and Numina datasets is similar to GRPO. (c, d) Using token level normalization with PL and NL results in improved training stability.

To explain the mystery behind GRPO's enhanced stability, we revert back to analyzing the role of advantages used in GRPO. Advantages reweigh the gradients for the $i^{th}$ rollout by multiplying them with the following quantity: $\tilde{A}_\pi(q, a^i) = \frac{r(a^i) - V(q)}{\mathbb{E}_{\tilde{a} \sim \pi_{\theta_{\text{old}}}(\tilde{a}|q)}(r(\tilde{a}) - V(q))^2}$. Here we will show that the norm of the gradients reduces when the model enters into a space close to either of the critical solutions $S_{PL}, S_{NL}$. If the model enters very close to $C_{NL}$, it will start generating very high entropy solutions, which will likely become incorrect. Or else if the model comes very close to $C_{PL}$, it will completely loose diversity in its generation (i.e. all rollouts will become same). In both the cases, the gradients will become zero in expectation for the incorrect answers or the same solutions respectively. However, for the few correct or diverse solutions (with different correctness as compared to other same solutions), the gradients will become relatively very large in magnitude. This will prevent the model to traverse further into the direction of the closeby critical solutions. The use of advantages therefore helps in stabilizing GRPO and prevents it from collapsing into either of the two bad critical solutions. We summarize this finding below:

> **Takeaway 3**
>
> The use of advantages in GRPO aids the optimization process by preventing it from converging to the critical solutions for $L_{\text{PL}}$ and $L_{\text{NL}}$, thereby stabilizing training.

## 5 IMPROVING STABILITY

Having analyzed the key design choices in GRPO's objective function, there is yet another recent development related to modifying loss normalization. We dive deeper into this design choice here. DAPO (Zhou et al., 2025) and Dr GRPO (Liu et al., 2025) highlighted that dividing the gradients by the sequence length of the rollouts introduces a length bias, which could lead to training instabilities. This makes the model prefer outputting longer reasoning chains, which leads to decreased training efficiency and also introduces training instabilities. In this section, we provide a deeper explanation about how introducing token level normalization as proposed in these works helps improve training stability. For this, we first analyze $L_{CL}$ with token level normalization, which yields the following objective function:

$$L_{\text{TNCL}}(\theta, q) = -\frac{1}{(|\mathcal{A}^+| + \mathcal{A}^+|)T}\Big(\sum_{a \in \mathcal{A}^+}\sum_{t=1}^{|a|}\frac{\pi_\theta(a_t|s_t)}{\pi_{\theta_{\text{old}}}(a_t|s_t)} - \sum_{a \in \mathcal{A}^-}\sum_{t=1}^{|a|}\frac{\pi_\theta(a_t|s_t)}{\pi_{\theta_{\text{old}}}(a_t|s_t)}\Big) \qquad (12)$$

where $T$ represents the maximum possible sequence length. We compare TNCL, GRPO, and CL for on-policy setting in Fig. 6 (a, b) and Table 1. Improved stability of using token normalization with CL is clearly evident. Note that although, TNCL is not as stable as GRPO, but in cases where collapse is observed, it is significantly delayed as compared to CL. We also observe delayed collapse when minimizing $L_{\text{PL}}$ and $L_{\text{NL}}$ individually but with token level normalization (See Fig. 6 (c, d)). To understand the cause behind the enhanced stability, we dive deeper into characterizing the critical solutions $S_{PL}$ and $S_{NL}$, and find that they change their form on using token level normalization:

$$S'_{PL} : \pi_\theta(a_t|s_t) = 1 \forall t \le |a|, |a| + |q| = T, \forall a \in \mathcal{A}^+ \qquad (13)$$

$$S'_{NL} : \pi_\theta(a_t|s_t) = \frac{1}{|V|}\forall t \le |a|, |a| = 1, \forall a \in \mathcal{A}^- \qquad (14)$$

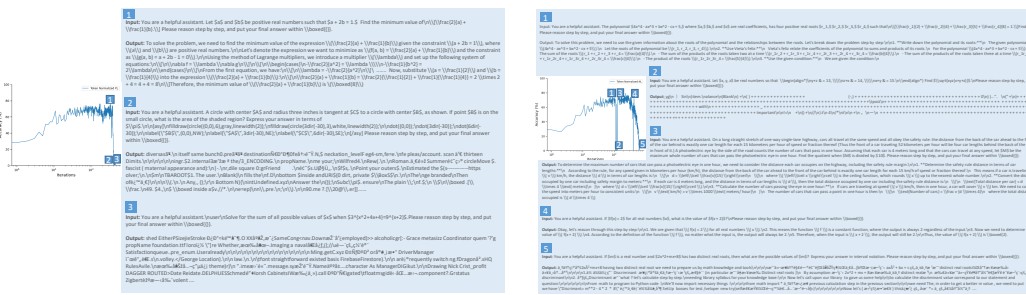

(a) $L_{\mathrm{PL}}$ (token level normalization)  (b) $L_{\mathrm{NL}}$ (token level normalization)

Figure 7: **Analysis of rollouts on SimpleRL for $L_{\mathrm{PL}}$ and $L_{\mathrm{NL}}$** objectives with token level normalization using Qwen2.5-7B model in on-policy setting. Using token level normalization changes the nature of rollouts at the time of collapse. As predicted by Eq. 13, 14, using $L_{PL}$ (a) increases length of rollouts at the time of collapse, whereas it decreases on using $L_{NL}$ (b). This is in sharp contrast to the observations predicted without the presence of token normalization in Fig. 4

where $|V|$ represents the size of the vocabulary. Refer to App. K for more details. Clearly, for the solutions $S'_{PL}$ and $S'_{NL}$, the length of rollouts is in contrast with the desired distribution of $\pi_\theta$. A high entropy, uniform distribution of $\pi_\theta$ naturally prefers longer outputs but the desired length for the optimal loss is unity. On the other hand, skewed distribution in case of $S'_{PL}$ will prefer shorter output length, but the desired solution requires longer outputs. This conflict indeed makes it difficult for the model to converge to $S'_{PL}$ and $S'_{NL}$, which in-turn leads to improved stability. This is clearly evident by looking at the rollouts generated on using token normalization with PL, NL, and CL in Fig. 7(a), 7(b), and 18 respectively. The nature of rollouts has also changed when compared to not using normalization (See Fig. 4(a), 7, and 16). These results provide a new lens explaining the increased effectiveness of using token level normalization, which has recently gained traction (Yu et al., 2025; Liu et al., 2025; Yue et al., 2025b).

> **Takeaway 4**
>
> Token level normalization creates a conflict between the properties of the critical solutions corresponding to minimization and maximization of entropy and the length of rollouts. This prevents the model from converging to them, thereby enhancing training stbility.

## 6 CONCLUSION

In this work, we unwrap GRPO's objective function and through rigorous experiments on multiple models and datasets, we discover instability and collapses in the training algorithms proposed in recent works in on-policy setting and further present the reasons for it. Next, we demonstrate how the advantages used in GRPO help in overcoming this instability, making it stable in on-policy settings. We note that due to limited access to compute our results are focused on the academically feasible setting of models up to a size of 8B and open-source datasets. It would be interesting to analyze how clipping helps induce stability, especially in cases where there is no policy improvement guarantee.

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

# A RELATED WORKS

## A.1 LLM REASONING

Explicit efforts towards improving the reasoning abilities of language models via RLVR started primarily with the advent of DeepSeek R1 (DeepSeek-AI, 2025), which demonstrated that utilizing verifiable outcome reward models (ORMs) could lead to significant improvements in performance of language models on tasks like maths and coding. Prior to this, GPT4 O1 (Jaech et al., 2024) also attracted a lot of interest within the community, however until DeepSeek R1, many were skeptical about ORMs. Given the demonstrated effectiveness of ORMs by DeepSeek R1, we consider the same sparse reward setting in this work. Another distinct characteristic highlighted by DeepSeek R1 was the emergence of long chain of thoughts with cognitive behaviors like self correction when performing RLVR fine-tuning. It was further argued that emergence of such behaviors helped in improving the reasoning abilities of the model Gandhi et al. (2025); Setlur et al. (2025). Gandhi et al. (2025) also demonstrated similar behaviors being learned in smaller models when being trained on more synthetic tasks like countdown. Some of the recent works: Setlur et al. (2025); He et al. (2025); Wang et al. (2025a); Qu et al. (2025) have argued that improvements observed on performing RLVR are driven by two mechanisms: 1) sharpening of existing skills 2) chaining of basic skills existing in the pretrained model. While the former helps in exploitation, the latter aids in exploration. However, the extent of contribution of these mechanisms towards improving model's performance remains unclear as highlighted by some recent works: Wu et al. (2025); Yue et al. (2025a); Zhao et al. (2025a). These works argue against the chaining hypothesis, by highlighting reduction in pass@$n$ performance of the RL tuned model when compared with the base model. Nevertheless, since DeepSeek R1 used GRPO as its preference learning objective function, GRPO has been believed as one of the major contributor behind the emergence of cognitive behaviors and improved reasoning abilities. However, the motivation behind the origins and several design choices of GRPO is still obscure. This motivates us to deeply understand the inner working and motivation behind different design choices of GRPO.

## A.2 ALGORITHMS FOR RL REASONING

Motivated by the effectiveness of GRPO, many follow up works have further proposed simplified versions of GRPO, often leading to similar or slightly improved performance (Zheng et al., 2025; Zhu et al., 2025; Zhao et al., 2025c; MiniMax, 2025; Chen et al., 2025a; Ahmadian et al., 2024; Xiong et al., 2025; Samineni et al., 2025). Reinforce++ (Xiong et al., 2025) demonstrated that training only on the correct rollouts gives very similar performance when compared with GRPO, thereby proposing a simpler alternative to GRPO. On the other hand, Zhu et al. (2025) demonstrated that minimizing the likelihood only on the incorrect rollouts gives similar performance as compared to GRPO while improving model's output diversity. Additionally, Samineni et al. (2025) demonstrated that a simple combination of the positive and negative losses as described above, also leads to performance similar to GRPO. Similarly, MiniMax (2025); Ahmadian et al. (2024) modify the clipping mechanisms in GRPO, thus leading to the same objective function as GRPO in the on-policy setting. Zheng et al. (2025); Zhao et al. (2025c) modify the GRPO's objective function by considering importance sampling at sequence level, which also leads to the same objective function as GRPO in on-policy setting. All these methods aim to simplify GRPO's objective function and they have successfully demonstrated their effectiveness on certain datasets and base models. These results clearly question the lack of motivation about different design choices in GRPO's objective function.

Recently a plethora of works have also tried using different reward functions Prabhudesai et al. (2025); Kang et al. (2025); Zhao et al. (2025b); Shafayat et al. (2025); Shao et al. (2025); Aggarwal & Welleck (2025), while utilizing GRPO for optimization. A few of them including Shao et al. (2025), have recently demonstrated improvements on utilizing GRPO with spurious rewards like format based rewards, thereby highlighting the importance of formatting. Surprisingly these improvements continue to persists even on using random rewards Shao et al. (2025). Prabhudesai et al. (2025) highlighted that using negative Shannon entropy as the reward could also lead to improved performance. On the other hand, contrastingly Wang et al. (2025b) demonstrated that maximization of entropy could also lead to improved performance. These results therefore have created confusion within the community, highlighting a need to understand the optimization landscape of GRPO and related simplified objective functions.

Another line of work has shown that the widely adopted preference learning method PPO gives sub-optimal performance when used for RLVR (Kazemnejad et al., 2024; Xiong et al., 2025). This

additionally makes it important to understand the reasons which make GRPO superior to PPO in the reasoning landscape. Now we will describe the background essential to understand the origins of GRPO and dive further into understanding the motivation and utility of different design choices of GRPO.

## B  BACKGROUND

In this section, we will discuss the background and motivation behind different preference optimization methods namely REINFORCRCE (Williams, 1992; Sutton et al., 1999), PPO (Schulman et al., 2017b), and GRPO (Shao et al., 2024). Let us consider an episodic MDP given by the tuple $(S, A, P, r, \gamma)$, where $S$ is a set of states, $A_s$ is a set of actions allowed for a given state $s$, and $A$ is the set of $A_s$. Here the policy is parametrized by $\theta$ and defined as $\pi_\theta : R^d \to [0, 1]^v$, where $v$ represents the number of classes and $d$ is the dimension of the input. Let each episode start at the state $s_0$. Denote by $P : S \times A \times S \to R$ the transition probability matrix, by $r_t$ the reward given by the environment at time stamp $t$, and by $\gamma$ the discounting factor. Let us assume that the process is episodic and always starts from a state $s_0$. Given this, we can now define the value function for our policy $\pi_\theta$:

$$v_{\pi_\theta}(s_0) = \sum_a \pi(a|s_0) q_\pi(s, a) \tag{15}$$

Here $q_\pi(s, a)$ represents the Q-value function defined as $E_\pi[G_t|a, s]$, where $G_t = r_t + \gamma r_{t+1} + ... + \gamma^n r_{t+n}$, where $s_{t+n}$ is the terminal state. The objective here is the maximize the value function for our policy, while updating the policy. Using the policy gradient theorem Sutton et al. (1999), we have the following:

$$\nabla_\theta v_{\pi_\theta} \propto \sum_s \mu(s) \sum_a q_\pi(s, a) \nabla_\theta \pi_\theta(a|s) \tag{16}$$

Here, $\mu(s)$ represents the relative frequency of visiting the state $s$ by the agent on following the policy $\pi_\theta$. We can further simplify the above expression in the following way:

$$\sum_s \mu(s) \sum_a q_\pi(s, a) \nabla \pi_\theta(a|s) = E_s[\sum_a q_\pi(s, a) \nabla \pi_\theta(a|s)] = E_{s,a \sim \pi_\theta}[q_{\pi_\theta}(s, a) \nabla \log(\pi_\theta(a|s))] \tag{17}$$

$$\theta_{t+1} = \theta_t + \alpha E_{s,a \sim \pi_\theta}[q_{\pi_\theta}(s, a) \nabla \log(\pi_\theta(a|s))] \tag{18}$$

Here $\alpha$ represents the learning rate. In practice, monte-carlo sampling is performed to get an unbiased estimate of the expectation. It is trivial to show that we can rewrite Eq. 18 as $\theta_{t+1} = \theta_t + \alpha E_{s,a}[(q_\pi(s, a) - v_\pi(s)) \nabla \log(\pi_\theta(a|s))]$ because $\sum_a v_\pi(s) \nabla \pi_\theta(a|s)] = 0$ and therefore subtracting this term from Eq. 16 won't add any bias. Let us call $q_\pi(s, a) - v_\pi(s)$ as the advantage function denoted by $A_\pi(s, a)$. Thus we get the following update:

$$\theta_{t+1} = \theta_t + \alpha \mathop{\mathbb{E}}_{s,a \sim \pi_\theta}[A_\pi(s, a) \nabla \log[\pi_\theta(a|s)]] \tag{19}$$

The monte carlo approximation of the above equation for a single rollout gives us the standard reinforce algorithm, which is given by

$$\theta_{t+1} = \theta_t + \frac{\alpha}{|a|} \sum_{k=1}^{|a|}[A_\pi(s_k, a_k) \nabla \log[\pi_\theta(a_k|s_k)]] \tag{20}$$

In contrast to reinforce, the motivation behind deriving PPO is a bit different. Due to limited scope of this work, we discuss the high level idea behind PPO's derivation below. PPO is motivated from Trust Region Policy Optimization (TRPO) (Schulman et al., 2017a), which tries to maximize a lower bound on the true policy's value function while guaranteeing that the policy improves in every iteration. To guarantee this, the approximated policy needs to remain within some proximity of the original policy. This leads to a constrained optimization problem, where KL divergence between the true and approximated policy is constrained. However, in practice the strict constrained optimization problem is difficult to optimize and therefore PPO makes an approximation of using clipping in order to enforce the closeness constraint. However, this results in the loss of any improvement guarantees and the selection of clipping hyperparameters are left to the users based on empirical evidence. Thus

the *utility of clipping in case of PPO remains unclear*. The objective function of PPO is given as:

$$J(\theta, q) = \mathbb{E}_{a \sim \pi_{\theta_{\text{old}}}(a|q)} \frac{1}{|a|} \sum_{t=1}^{|a|} \min\big[\frac{\pi_\theta(a_t|s_t)}{\pi_{\theta_{\text{old}}}(a_t|s_t)} A_\pi(s_t, a_t), \text{clip}(\frac{\pi_\theta(a_t|s_t)}{\pi_{\theta_{\text{old}}}(a_t|s_t)}, 1-\epsilon, 1+\epsilon) A_\pi(s_t, a_t)\big]$$

(21)

where $\pi_\theta$ and $\pi_{\theta_{\text{old}}}$ represent the current and the old policy utilized for sampling the rollouts. $q$ represents the input prompt and $a$ represents a single rollout generated by the model. PPO utilizes generalized advantages which are motivated from temporal difference learning Sutton (1988), which is based on unbiased approximation of the q-value functions using value function of future states. The generalized advantages used in PPO are given by

$$\hat{A}_\pi(s_t, a_t) = -v_\pi(s_t) + r_t + \gamma r_{t+1} + \gamma^2 r_{t+2} + ... + \gamma^{T-t-1} r_{t+1} + \gamma^{T-t} v_\pi(s_T)$$
$$= \delta_t + \gamma\lambda\delta_{t+1} + ... + (\gamma\lambda)^{T-t-1}\delta_{T-1} \quad \text{where} \quad \delta_t = r_t + \gamma V(s_{t+1}) - V(s_t), \ \lambda = 1 \quad (22)$$

PPO uses a value network to predict the value functions. This value network is in-turn trained using the ground truth rewards obtained for the rollouts generated during training.

GRPO (Shao et al., 2024) has followed the footsteps of PPO, and simply replaced the calculation of advantages via monte carlo samplings. However, there are several approximations which undergo here, which we describe in Sec. 2.

## C  OFF-POLICY FOR PPO, UNSUPERVISED RL, AND NOISY REWARDS

Building on our preliminary evidence supporting the enhanced stability of off-policy training in Fig. 3, we further investigate the robustness of this observation on other RLVR methods including CL, PPO (Schulman et al., 2017b), unsupervised RL (Prabhudesai et al., 2025), and GRPO with noisy rewards. As shown in Fig. 5 (e), using off-policy setting with clipping and importance sampling makes the training of combined loss stable, while it remains unstable in on-policy setting. Similarly, as demonstrated in Fig. 8 (b), we find that training Qwen2.5-7B with unsupervised RL by simply minimizing entropy as proposed in Prabhudesai et al. (2025) collapses quickly in on-policy setting. These results corroborate with our findings about Zhu et al. (2025) and Xiong et al. (2025).

Next, we craft an adversarial setting, where we utilize incorrect rewards for 25% of the rollouts. This leads to calculation of incorrect advantages in GRPO, thereby prompting instability in training as the gradients don't become zero even when all the outputs are correct or incorrect. We observe GRPO to remain in on-policy setting, as well off-policy setting with clipping (See Fig. 8 (a)). Finally, we investigate the stability of the popularly used preference learning method PPO (Schulman et al., 2017b) in RLVR setting. As shown in Fig. 8 (c, d), PPO collapses in on-policy setting for a few datasets and models, but it remains stable in off-policy setting (more details in Appendix E).

The above results highlight an imperative role played by clipping in stabilizing the training on varying datasets, models, and training algorithms. However, GRPO still remains stable in on-policy setting. We note that we do not have a complete understanding about the mechanism which helps clipping stabilize the training. However, preliminary results indicate that clipping reduces the norm of the gradients as shown in Fig. 14. We hypothesize that this happens because clipping makes the gradients zero precisely when the model becomes extremely confident or extremely uncertain about its prediction, thereby preventing large deviations from the base model used for sampling rollouts. This prevents updates which could lead to collapse or explosion of model's entropy. Understanding the mechanisms used by clipping to induce training stability is an interesting future direction. We now highlight the main takeaway from this discussion below.

## D  INSTABILITY IN OFF-POLICY SETTING

We observed in the main paper in Fig. 9, that instability of training increases on training in off-policy setting. To understand this in detail, we revert back to Eq. 9, and find that a small value of $\pi_\theta$ should lead to larger norm of gradients, thereby expediting collapse. To confirm that this is indeed the case, we analyze the evolution of difference between the probability of sampling the ground truth tokens from the old policy $\pi_{\theta_{\text{old}}}$ and the current policy $\pi_\theta$ (denoted by $\Delta\pi_\theta$) in Fig. 15. We observe that $\Delta\pi_\theta$ increases over the course of training, and becomes significantly large at the time of collapse.

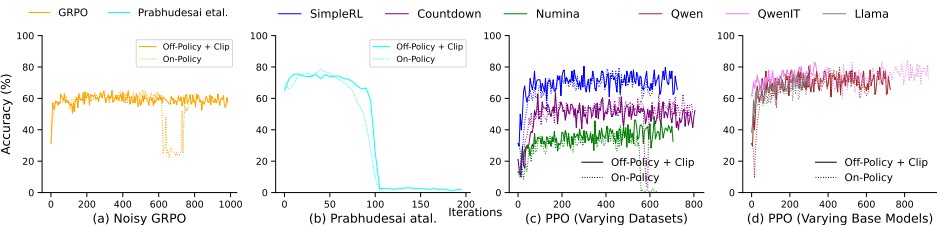

Figure 8: **Instability of PPO, Noisy GRPO, and Prabhudesai et al. (2025) in on-policy setting:** (a) Instability in training is observed for GRPO when using noisy rewards in on-policy setting. However, the training becomes stable in off-policy setting on using clipping. (b) Prabhudesai et al. (2025) undergoes collapse in on-policy as well as off-policy setting. (c) PPO undergoes collapses on training using Numina Hard and experiences instability when training on Countdown. But, its training remains stable on SimpleRL across different models (d).

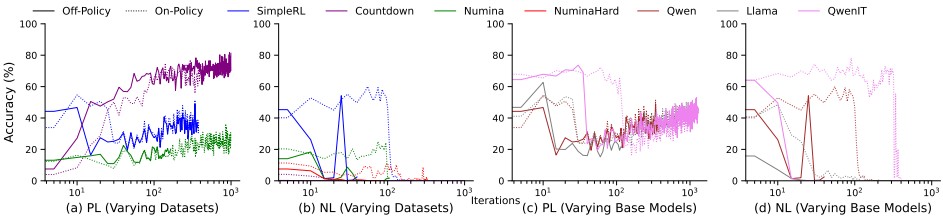

Figure 9: **Off-Policy training without clipping and without importance sampling:** Comparison between the evolution of training accuracy for PL (Off-Policy) and PL (On-Policy) (a, c) and NL (Off-Policy) and NL (On-Policy) (b, d). Off-Policy training expedites the onset of collapses.

## E  CASE STUDY ON PPO

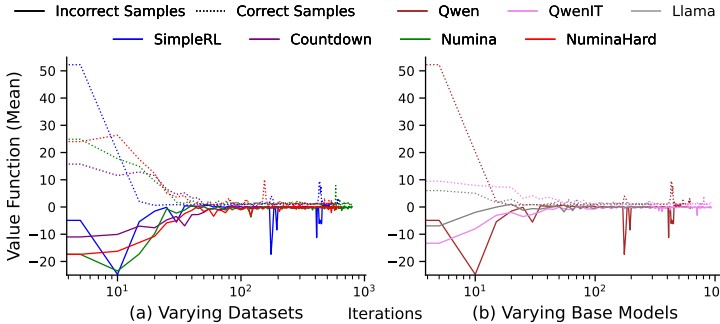

Figure 10: **Case study on PPO:** The mean of the value functions predicted by the value model shows clear separation between correct and incorrect samples, where the incorrect samples often get negative values, and the correct ones receive positive ones. This shows that the value estimates predicted by the value network often deviate significantly from the true estimates.

We note that the instability of PPO demonstrated in on-policy setting in Section C could be of independent interest, as PPO is one of the most popular preference learning method widely used across different domains including RLVR, RLHF, etc. To understand the root cause for the observed instability, we dive deeper into analyzing the estimates of value function calculated by the value network used in PPO. Surprisingly, we find that the value network predicts negative value functions in case of incorrect rollouts (See Fig. 10). Note that this is mathematically not possible, as the value function should be greater than zero when using zero reward for incorrect rollouts and one for the correct ones. We note that similar inconsistencies were also noted previously in Kazemnejad et al. (2024) which motivated them to utilize monte carlo rollouts for calculating advantages instead of a value network. These results indicate that although PPO seems quite mathematically principled,

but in practice the estimators for advantages have large errors from the true estimates. This leads to collapses in on-policy setting as observed in Fig. 8 (c, d). This finding is summarized below.

---

**Takeaway 5**

PPO in RLVR setting without the use of KL divergence is susceptible to training instabilities often leading to model collapses. This is due to high errors associated with the empirical estimators of value functions.

---

Next, we provide a preliminary study to overcome the training instabilities observed so far.

## F  DISCUSSION

We note that the although the analysis presented in this work is focused on RLVR, our results are expected to hold across different applications of preference learning, where algorithms like GRPO are being used. This also includes RLHF, which is popularly used for aligning the language models as per human preferences. One example particularly interesting is SimPO (Meng et al., 2024) which was proposed as a simplified version of DPO (Rafailov et al., 2023). We can show that SimPO behaves like reweighted maximum likelihood estimation (MLE) in cases where the confidence of the policy on the incorrect samples is much larger than that on the correct samples. This is in fact the regime where maximum learning would occur. More details are given below:

The optimization function of SimPO is given by the following

$$L_{SimPO}(\pi_\theta) = -E_{(x,y_w,y_l)\sim D}[log\ \sigma(\frac{\beta}{|y_w|}log\ \pi_\theta(y_w|x) - \frac{\beta}{|y_l|}log\ \pi_\theta(y_l|x) - \gamma)] \quad (23)$$

where $y_w, y_l$ represents the preferred and the less preferred outputs for the input prompt given by $x$. Gradients for the above objective function are given as:

$$\nabla_\theta L_{SimPO}(\pi_\theta) = E_{(x,y_w,y_l)\sim D}[\nabla_\theta log(1 + e^{-\frac{\beta}{|y_w|}log\ \pi_\theta(y_w|x) + \frac{\beta}{|y_l|}log\ \pi_\theta(y_l|x) + \gamma})] \quad (24)$$

Clearly, if $-\frac{\beta}{|y_w|}log\ \pi_\theta(y_w|x) + \frac{\beta}{|y_l|}log\ \pi_\theta(y_l|x) + \gamma >> 0$ then we have the following approximation

$$log(1 + e^{-\frac{\beta}{|y_w|}log\ \pi_\theta(y_w|x) + \frac{\beta}{|y_l|}log\ \pi_\theta(y_l|x) + \gamma})] \approx -\frac{\beta}{|y_w|}log\ \pi_\theta(y_w|x) + \frac{\beta}{|y_l|}log\ \pi_\theta(y_l|x) + \gamma \quad (25)$$

Thus, we get the following

$$\nabla_\theta L_{SimPO}(\pi_\theta) = E_{(x,y_w,y_l)\sim D}\ \nabla_\theta[-\frac{\beta}{|y_w|}log\ \pi_\theta(y_w|x) + \frac{\beta}{|y_l|}log\ \pi_\theta(y_l|x) + \gamma] \quad (26)$$

The above equation is simply some reweighted version of MLE.

Similarly as shown in Fig.10, we observe that PPO ends up learning positive value functions for correct rollouts and negative for the incorrect ones, thereby resulting in an objective function very close to simple reweighted iterated MLE. The demonstrated similarity of different preference learning methods with simple maximum likelihood estimation makes us question if we are really investing correctly in terms of algorithmic designs for learning preferences. Here is one such way that could help develop more principled preference learning algorithms:

**Performing reweighted iterated MLE at token level:** PPO (Schulman et al., 2017b) and Reinforce (Sutton et al., 1999) can be considered as doing token level MLE. However due to several errors in PPO (See Sec. E) and approximations made in Reinforce, we end up in a regime very close to sample level reweighting. A naive solution to perform token level reweighting could be to generate multiple monte-carlo rollouts at each state of the MDP. However, this could be quite expensive in terms of compute. Moreover, defining a good MDP need not define a state as a single token prediction. Therefore, definition of an MDP with appropriate number and position of states is important. A naive solution could be to prompt an LLM for this, but there could be better ways. Utilizing process reward models could also be helpful in minimizing the compute associated with rollouts, however training a good process reward model is challenging in itself (Lightman et al., 2023).

**Incorporating preference based learning during pre-training:** An alternative way to enable models learn human like preferences could be to incorporate preference based training during pretraining itself by designing specialized datasets and training algorithms. In such a case, the effectiveness of existing post-training methods like PPO and GRPO would increase, even if they end up doing simple reweighted MLE.

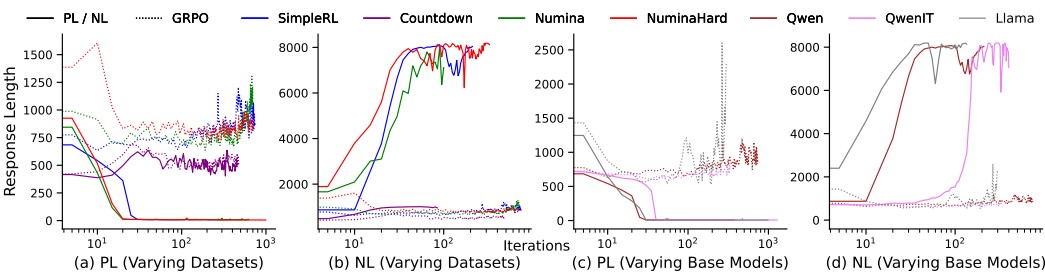

(a) PL (Varying Datasets)     (b) NL (Varying Datasets)     (c) PL (Varying Base Models)     (d) NL (Varying Base Models)

Figure 11: **Analysis of response length over training iterations in on-policy setting:** Using $L_{\mathrm{PL}}$ collapses the output's length, while utilizing $L_{\mathrm{NL}}$ explodes it when compared to optimizing GRPO.

## G    REASONING IN BANDITS SETTING

Here, we try to derive GRPO's objective function in the setting of contextual bandits. Considering a single state MDP we will get the following gradient update for GRPO.

$$\theta_{t+1} = \theta_t + \alpha \frac{1}{K} \sum_{i=1}^{K} A_\pi(q, a^i) \nabla_{\theta_t} [\prod_{t=1}^{|a^i|} f(\pi_{\theta_{\mathrm{old}}}(a_t^i|s_t, ), \pi_\theta(a_t^i|s_t), \epsilon)],$$

$$where \ \ A_\pi(s_t, a_t^i) = \frac{r - \frac{1}{K} \sum_{k=1}^{|K|} r_k}{std(r_k)}$$

$$f(\pi_{\theta_{\mathrm{old}}}(a_t^i|s_t)), \pi_\theta(a_t^i|s_t), \epsilon) = \min \left[ \max \left[ \frac{\prod_{t=1}^{|a^i|} \pi_\theta(a_t^i|s_t)}{\prod_{t=1}^{|a^i|} \pi_{\theta_{\mathrm{old}}}(a_t^i|s_t)}, 1 - \epsilon \right], 1 + \epsilon \right] \quad (27)$$

Clearly, Eq. 27 is different from Eq. 3, which means that GRPO cannot be considered as operating in a bandit setting. However, it would be interesting to analyze how much of a difference analyzing GRPO (Shao et al., 2024) and DAPO (Zhou et al., 2025) in an MDP vs. bandit setting would create. This has been recently explored in Zheng et al. (2025); Zhao et al. (2025c), which demonstrate improved stability and performance as compared to GRPO.

## H    GRADIENT UPDATES IN CASE OF PPO

Using PPO in on-policy setting we get the following:

$$\theta_{t+1} = \theta_t + \frac{\alpha}{|a_t|} \sum_{t=1}^{|a|} A_{t,\pi}(s_t, a_t) \frac{\nabla \pi_{\theta_t}(a_t|s_t, \theta)}{\pi_{\theta_{\mathrm{old}}}(a_t|s_t, \theta)}$$

where $A_{t,\pi}(s_t, a_t) = -v_{\pi_{\phi_t}}(s_t) + r_t + \gamma r_{t+1} + ... \gamma^{T-t} v_{\pi_{\phi_t}}(s_T)$ and $\phi_{t+1} = \phi_t - \frac{\beta}{K} \sum_{i=1}^{K} (r_i - v_{\pi_\phi})$

$$(28)$$

$\phi_t$ corresponds to the $t^{th}$ time step update of the value network. Since $v_{\pi_\phi}$ is calculated for each state, we can consider PPO as performing reweighted iterated MLE but with reweighting at token level.

## I    ADDITIONAL RESULTS

In this section, we present additional analysis providing a more detailed investigation on the results in the main paper. First, we analyze the evolution of the average response length over the course of training in on-policy setting in Fig. 11. We find that on optimizing $L_{\mathrm{NL}}$ the average length of model's outputs increases, while it decreases on optimizing $L_{\mathrm{PL}}$. Similarly, as shown in Fig. 12 the entropy of model's outputs increases on using $L_{\mathrm{NL}}$ and decreases on using $L_{\mathrm{PL}}$. Moreover the sudden jumps in entropy are close to the timestamps where the model collapses. Similarly, on analyzing the evolution

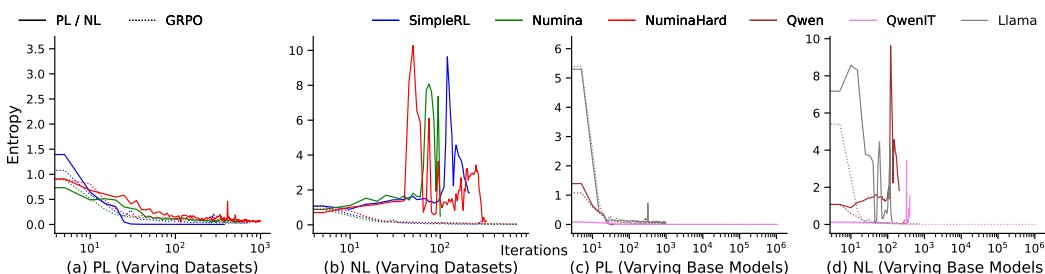

Figure 12: **Analysis of output's entropy over training iterations in on-policy setting:** Using $L_{\mathrm{PL}}$ collapses the output's entropy, while utilizing $L_{\mathrm{NL}}$ explodes it when compared to optimizing GRPO.

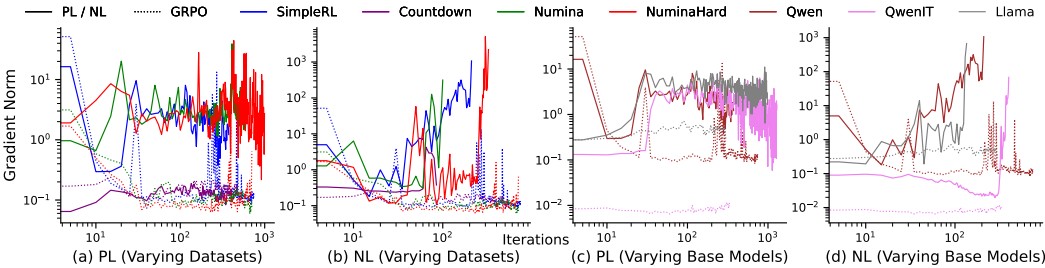

Figure 13: **Analysis of gradient's norm over training iterations in on-policy setting:** Generally the gradient norm is higher when optimizing $L_{\mathrm{PL}}$ and $L_{\mathrm{NL}}$ when compared with GRPO. The norm becomes especially high at the time of collapse and generally remains high thereafter.

of model's grdient norm over the course of training in Fig. 13, we find that the norm of the gradient increases drastically close of the point where the model collapses and training instability is observed.

Next, we compare the evolution of model's gradient norm between off-policy and on-policy training, where clipping is utilized in off-policy training. As shown in Fig. 14, using clipping leads to reduction in gradient's norm which in-turn leads to improved training stability. To further understand the reason behind this, we analyze how the average difference between the probability of the current and the old policy changes over the course of training in Fig. 14. We observe that utilizing clipping reduces this difference, which results in lower norm of gradients.

We further analyze the evolution of the average value functions predicted by the value network on using PPO in Fig. 10. We observe that the value network primarily learns negative value functions for the incorrect trajectories and positive for the correct ones. Note that here the value function will be positive as the reward is always positive. Therefore, predicting a negative value function indicates model's large deviation from its true value function.

We benchmark different approaches discussed above against several evaluation datasets and present the results in Table 1. We observe that while optimizing $L_{\mathrm{NL}}, L_{\mathrm{PL}}$, and $L_{CL}$ leads to suboptimal performance when compared with GRPO, on using token normalization, optimizing $L_{TNCL}$ leads to performance comparable with GRPO.

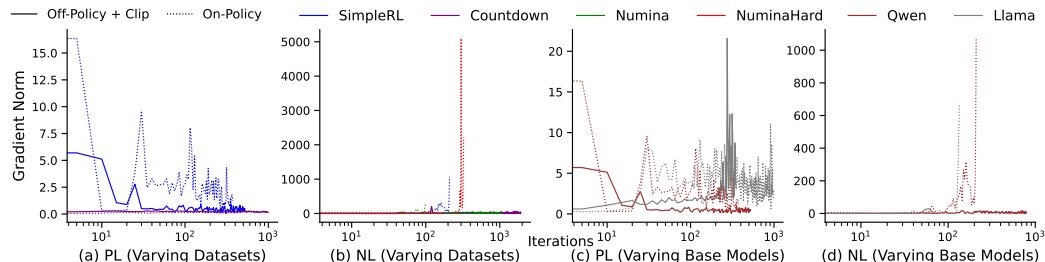

Figure 14: **Comparison of gradient's norm between on and off policy training when clipping is utilized.** Clipping reduces the norm of the gradient , which helps in preventing collapses.

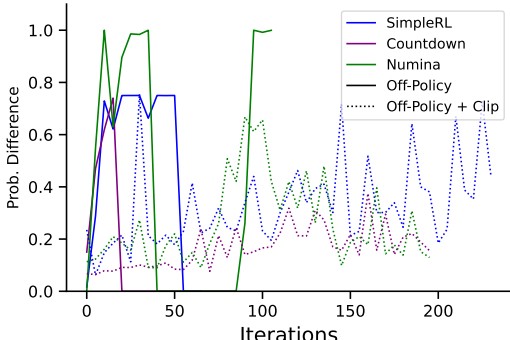

Figure 15: **Analysis of the difference in probability of sampling rollouts from the old and the current policy in off-policy setting.** The difference increases on using off-policy without clipping. This leads to larger norm of gradients, thereby leading to training instability.

Table 1: **Evaluation results of different objective functions in on-policy setting:** The results are suboptimal when optimizing $L_{PL}$, $L_{PL}$, and $L_{CL}$ as compared to GRPO. However on using token normalization, $L_{CL}$ performs similar to GRPO, thereby highlighting the effectiveness of token normalization.

| Method | Dataset | Model | GSM8K | Math-500 | College-Math | Gaokao-2023 | Minerva-Math | OlympiadBench |
|---|---|---|---|---|---|---|---|---|
| GRPO |  |  | 91.9 | 77.6 | 41.9 | 64.4 | 37.9 | 39.7 |
| PPO |  |  | 88.6 | 73 | 39.6 | 62.6 | 30.5 | 34.1 |
| NL | SimpleRL | Qwen | 0.1 | 0.2 | 0.1 | 0.8 | 0.7 | 0 |
| PL |  |  | 59.4 | 31.4 | 23.9 | 28.6 | 16.5 | 9.2 |
| CL |  |  | 1.4 | 0.4 | 0.5 | 4.7 | 0.7 | 1 |
| GRPO |  |  | 87.5 | 52.2 | 30.9 | 47.3 | 22.8 | 20 |
| PPO |  |  | 84.7 | 48.2 | 30.2 | 41.6 | 26.8 | 15.6 |
| NL | SimpleRL | Llama | 21.5 | 5.8 | 3.2 | 6.5 | 4 | 3.3 |
| PL |  |  | 17.1 | 14.8 | 11 | 17.4 | 9.2 | 5 |
| CL |  |  | 78.1 | 40 | 26.3 | 35.3 | 21.3 | 11.3 |
| GRPO |  |  | 90.8 | 77.6 | 40.5 | 61.3 | 43 | 38.8 |
| PPO |  |  | 91.7 | 77.8 | 42.3 | 64.2 | 40.1 | 40.4 |
| NL | SimpleRL | QwenIT | 2.2 | 2.6 | 1 | 4.4 | 1.8 | 1.3 |
| PL |  |  | 25.2 | 27.8 | 20.5 | 29.1 | 16.2 | 8.7 |
| CL |  |  | 77.2 | 59 | 32.5 | 50.6 | 29.4 | 28 |
| GRPO |  |  | 90.8 | 74.6 | 41.7 | 63.4 | 34.2 | 37.9 |
| NL | Numina-Hard | Qwen | 1.6 | 1.8 | 1 | 2.9 | 1.1 | 1.2 |
| PL |  |  | 23.7 | 22.8 | 17.9 | 28.3 | 15.1 | 7 |
| CL |  |  | 0.4 | 1 | 1.4 | 1.8 | 0.4 | 0.1 |
| GRPO |  |  | 91.1 | 77 | 45.6 | 64.4 | 38.6 | 39.9 |
| NL | Numina | Qwen | 1.7 | 1.8 | 0.7 | 4.7 | 1.1 | 0.9 |
| PL |  |  | 25.2 | 24.4 | 20.8 | 26.8 | 16.2 | 7.9 |
| CL |  |  | 52.8 | 41.6 | 25.5 | 36.1 | 16.2 | 23.3 |
| GRPO (Token Norm.) |  |  | 91.9 | 70 | 40.7 | 59.7 | 37.9 | 33.8 |
| NL (Token Norm.) | SimpleRL | Qwen | 56.8 | 42.2 | 23.1 | 37.7 | 7 | 15.1 |
| PL (Token Norm.) |  |  | 23.4 | 21.2 | 11.3 | 22.6 | 6.2 | 6.8 |
| TNCL |  |  | 91.8 | 77.8 | 41 | 65.2 | 40.1 | 37.6 |

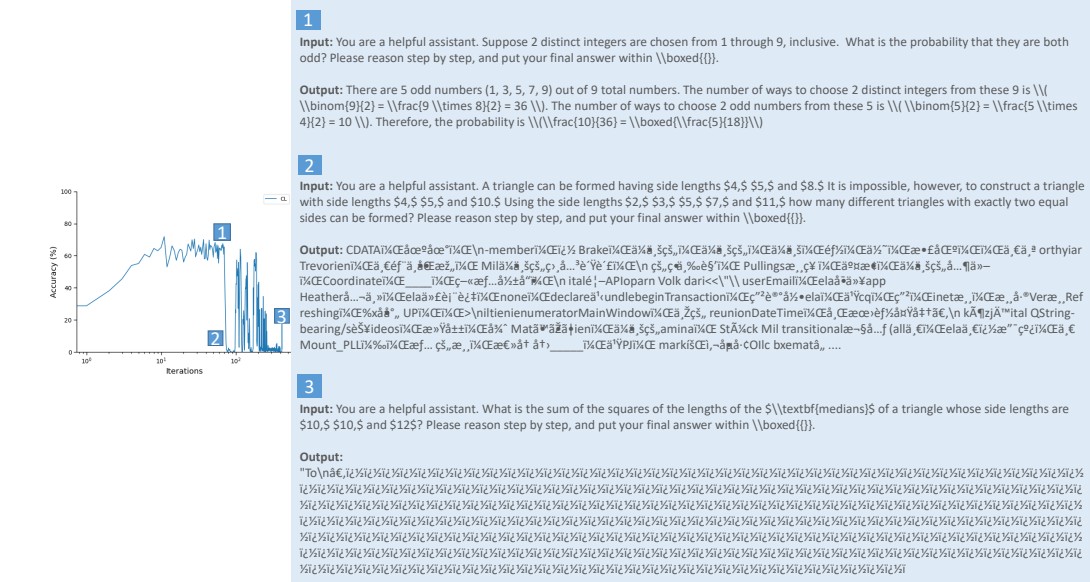

**Figure 16: Analysis of rollouts for Qwen2.5-7B trained using $L_{CL}$ in on-policy setting on SimpleRL.**

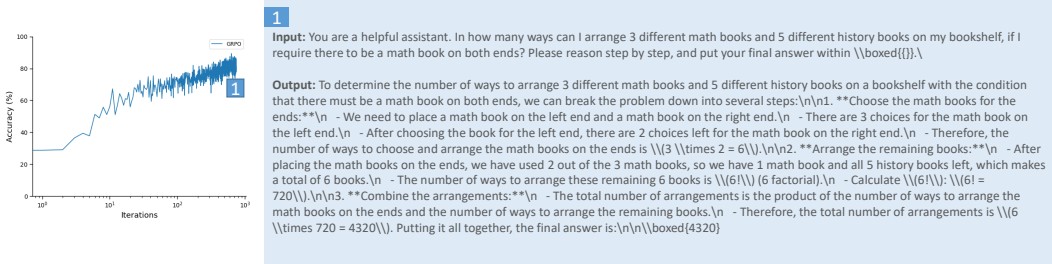

**Figure 17: Analysis of rollouts for Qwen2.5-7B trained using GRPO in on-policy setting on SimpleRL.**

## J  ANALYSIS OF ROLLOUTS

We investigate the rollouts corresponding to different objective functions (NL, PL, and CL), alongwith GRPO and their token normalized version in Fig. 4, 16, 4(a), 18, 7(b), and 7(a). We observe that on optimizing for $L_{\mathrm{NL}}$ and $L_{CL}$ at the time of collapse, the model learns to output random tokens. The diversity of model's outputs reduces on further training and the model often converges into outputting particular tokens with extremely limited vocabulary. On the other hand on using $L_{\mathrm{PL}}$ the model learns the shortcut solution of outputting the final answer directly without any reasoning traces. Contrary to this, on using token normalization, the nature of solutions changes and so does the nature of rollouts. Now, on using $L_{\mathrm{PL}}$ with token level normalization, the model learns to output large length solutions, which become more of random at the time of collapse. Whereas optimizing $L_{\mathrm{NL}}$ results in shorter outputs.

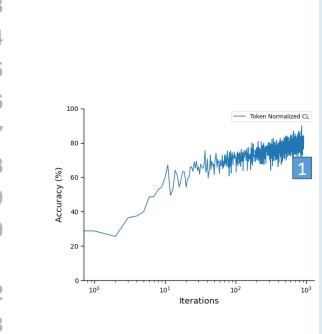

**Input:** You are a helpful assistant. Square $ABCD$ has side length $1$ unit. Points $E$ and $F$ are on sides $AB$ and $CB$, respectively, with $AE = CF$. When the square is folded along the lines $DE$ and $DF$, sides $AD$ and $CD$ coincide and lie on diagonal $BD$. The length of segment $AE$ can be expressed in the form $\sqrt{k}-m$ units. What is the integer value of $k+m$? Please reason step by step, and put your final answer within \boxed{{}}.

**Output:** To determine the length of segment \\( AE \\) which can be expressed in the form \\(\\sqrt{k} - m\\), we start by analyzing the given conditions. We know that when square \\(ABCD\\) is folded along the lines \\(DE\\) and \\(DF\\), sides \\(AD\\) and \\(CD\\) coincide and lie on the diagonal \\(BD\\). This implies that points \\(A\\) and \\(C\\) are reflected to lie on \\(BD\\).\n\nLet \\(AE = CF = x\\). Since \\(ABCD\\) is a square with side length 1, we have \\(BE = 1 - x\\) and \\(BF = 1 - x\\). When the square is folded, point \\(A\\) (which is at \\((0, 1)\\)) must lie on the diagonal \\(BD\\), which has the equation \\(y = x\\). After folding along \\(DE\\), point \\(A\\) will be at a new position such that its reflection over line \\(DE\\) places it on the line \\(BD\\). The same logic applies to point \\(C\\) (which is at \\((1, 0)\\)).\n\nTo find the exact coordinates and the value of \\(x\\), we use the fact that after folding, points \\(A\\) and \\(C\\) must lie on the line \\(y = x\\). For point \\(A\\), its reflection over line \\(DE\\) must place it on the line \\(y = x\\). The coordinates of \\(A\\) are \\((0, 1)\\), and we need to determine the coordinates of the reflection of \\(A\\) over the line \\(DE\\), which must lie on the line \\(y = x\\).\n\nThe line \\(DE\\) can be determined using the coordinates of \\(D (0, 0)\\) and \\(E (x, 0)\\). The slope of \\(DE\\) is 0, so the fold along \\(DE\\) will symmetrically place \\(A\\) such its new position \\((x_1, y_1)\\) must satisfy \\(y_1 = x_1\\). Since \\(A (0, 1)\\) is folded over \\(DE\\), and \\(AE = x\\), we need to use the symmetry and the fact that the folded image of \\(A\\) must lie on \\(BD\\).\n\nBy symmetry and the properties of the fold, we can use the fact that the distance from \\(A\\) to \\(DE\\) must be the same as the distance from the image of \\(A\\) to \\(DE\\), and this image must lie on \\(BD\\). Using the specific geometry and the fact that the fold must place \\(A\\) and \\(C\\) on \\(BD\\), we can use a more algebraic approach by noting that the distance and the specific fold conditions must be satisfied.\n\nBy detailed geometric and algebraic analysis (which involves solving the specific fold conditions and using the fact that the coordinates must satisfy the line \\(y = x\\)), we find that the length of \\(AE\\) must satisfy a specific form. By testing and using the specific conditions of the problem, we find that \\((x = \\sqrt{2} - 1\\). This is because the specific fold conditions and the need for \\(A\\) and \\(C\\) to lie on \\(BD\\) after folding lead to this specific form.\n\nThus, we have \\(AE = \\sqrt{2} - 1\\), so \\(k = 2\\) and \\(m = 1\\), and the integer value of \\(k + m\\) is \\(\\boxed{3}\\)

Figure 18: **Analysis of rollouts for Qwen2.5-7B trained using $L_{TNCL}$ in on-policy setting on SimpleRL.**

## K    CHARACTERIZATION OF CRITICAL SOLUTIONS

Here we will first derive the global minimas corresponding to the objective function in Eq. 9. We wish to show that

$$S_{\mathrm{NL}} : \pi_\theta(a_t|s_t) = \frac{1}{|V|} \forall t \le |a|, \forall a \in \mathcal{A}^- \tag{29}$$

The optimization objective is given as follows:

$$L_{\mathrm{NL}}(\theta, q) = -\frac{1}{|\mathcal{A}^-|} \sum_{a \in \mathcal{A}^-} \frac{1}{|a|} \sum_{t=1}^{|a|} A^-(q, a), \text{ where } a \sim \pi_\theta(a|q), A^-(q, a) = 1$$

$$\implies \nabla_\theta L_{\mathrm{NL}} = -\frac{1}{|\mathcal{A}^-|} \sum_{a \in \mathcal{A}^-} \frac{1}{|a|} \sum_{t=1}^{|a|} \frac{\nabla_\theta \pi_\theta(a_t|s_t)}{\pi_\theta(a_t|s_t)} = 0, \text{ where } a \sim \pi_\theta(a|q)$$

$$\implies \sum_a \frac{\pi_\theta(a|q)}{|a|} \sum_{t=1}^{|a|} \frac{\nabla_\theta \pi_\theta(a_t|s_t)}{\pi_\theta(a_t|s_t)} = 0$$

$$\implies \sum_a \frac{\Pi_{k=1}^{|a|} \pi_\theta(a_k|s_k)}{|a|} \sum_{t=1}^{|a|} \frac{\nabla_\theta \pi_\theta(a_t|s_t)}{\pi_\theta(a_t|s_t)} = 0 \tag{30}$$

Note that the above problem might have several critical solutions. But in particular, we can characterize only a few of them.

Assuming that the uniform policy would lead to a fixed length of the output given by $F$ and the cardinality of $\mathcal{A}^-$ tends to infinity, we can now show that using a uniform policy will serve as a critical solution to the above equation.

$$\sum_a \frac{\Pi_{k=1}^{|a|} \pi_\theta(a_k|s_k)}{|a|} \sum_{t=1}^{|a|} \frac{\nabla_\theta \pi_\theta(a_t|s_t)}{\pi_\theta(a_t|s_t)} = \frac{1}{F|V|^{F-1}} \nabla_\theta \sum_a \sum_{t=1}^{|a|} \pi_\theta = \frac{1}{F|V|^{F-1}} \nabla_\theta c = 0 \tag{31}$$

where $c$ is some constant. Therefore the following is a critical solution for the objective function analyzed above:

$$\pi_\theta(a_t|s_t) = \frac{1}{|V|} \forall t \le |a|, \forall a \in \mathcal{A}^- \tag{32}$$

Note that the above scenario is true only when the cardinality of the set $\mathcal{A}^-$ is close to infinity. In practice, this won't be true and therefore we might observe large variance depending on the cardinality of $\mathcal{A}^-$.

Next, we wish to show that

$$S_{\text{PL}} : \pi_\theta(a_t|s_t) = 1 \forall t \leq |a|, \forall a \in \mathcal{A}^+ \tag{33}$$

*Proof.* For $L_{\text{PL}}$, we get the following

$$\sum_a \frac{\Pi_{k=1}^{|a|}\pi_\theta(a_k|s_k)}{|a|} \sum_{t=1}^{|a|} \frac{\nabla_\theta\pi_\theta(a_t|s_t)}{\pi_\theta(a_t|s_t)} = \sum_a \sum_{t=1}^{|a|} \frac{\Pi_{k=1}^{|a|}\pi_\theta(a_k|s_k)}{|a|\pi_\theta(a_t|s_t)}\nabla_\theta\pi_\theta(a_t|s_t) = 0 \tag{34}$$

Now we will show that $\nabla_\theta\pi_\theta(a_t|s_t) = 0$ if $\pi_\theta(a_t|s_t) = 1 \forall t \leq |a|, \forall a \in \mathcal{A}^+$

$$\nabla_\theta\pi_\theta(a_t|s_t) = [\frac{e^{x_{a_t}}(\sum_{i=1}^{|V|}e^{x_i} - e^{x_{a_t}})}{(\sum_{i=1}^{|V|}e^{x_i})^2}\nabla_\theta x_{a_t}, \{-\frac{e^{x_{a_t}}(e^{x_j})}{(\sum_{i=1}^{|V|}e^{x_i})^2}\nabla_\theta x_j\}_{j=1,j\neq a_t}^{|V|}] \tag{35}$$

where $x_{a_t}$ represents the logits corresponding to $a_t$ and $x_i$ represents the logits corresponding to $i^{th}$ token in the vocabulary. The above equation will yield a vector of zeroes iff $x_j << x_{a_t}, \forall j \neq a_t, \forall t$.

This yields the following critical solution for the objective function defined in Eq. 9.

$$\pi_\theta(a_t|s_t) = 1 \forall t \leq |a|, \forall a \in \mathcal{A}^+ \tag{36}$$

$\square$

Note that there would be infinitely many solutions following Eq. 32 and Eq. 36 as the length of the rollouts (i.e. $|a|$) is not a constant. Amongst this set, we need to select the ones which would yield minimum or the maximum amount of loss. Since in case of Eq. 9, and Eq. 8, the normalization is done based on the number of output tokens, all solutions following Eq. 32, will yield the same loss. Same is the case with Eq. 36.

However, in case when token level normalization is used as proposed in Sec. 5, all the policies following Eq. 32 and Eq. 36 won't yield same loss. In case of Eq. 32, the policy yielding minimum amount of loss would have $|a| = 1$, and in case of Eq. 36, the policy yielding maximum loss would have $|a| = T$. Therefore, using a different normalization can change the behavior of the model.

Here we will show that $S_{NL}$ and $S_{PL}$ are critical solutions of $L_{CL}$:

*Proof.*

$$L_{CL}(\theta, q) = \frac{|\mathcal{A}^+|L_{\text{PL}}(\theta, q) + |\mathcal{A}^-|L_{\text{NL}}(\theta, q)}{|\mathcal{A}|} \tag{37}$$

where $|\mathcal{A}| = |\mathcal{A}^+| + |\mathcal{A}^-|$.

$$L_{\text{PL}}(\theta, q) = \frac{-1}{|\mathcal{A}^+|}\sum_{a\in\mathcal{A}^+}\frac{1}{|a|}\sum_{t=1}^{|a|}\frac{\pi_\theta(a_t|s_t)}{\pi_{\theta_{\text{old}}}(a_t|s_t)} \tag{38}$$

$$L_{\text{NL}}(\theta, q) = \frac{1}{|\mathcal{A}^-|}\sum_{a\in\mathcal{A}^-}\frac{1}{|a|}\sum_{t=1}^{|a|}\frac{\pi_\theta(a_t|s_t)}{\pi_{\theta_{\text{old}}}(a_t|s_t)} \tag{39}$$

From Eq. 35, we know that $\nabla_\theta\pi_\theta(a_t|s_t) = 0$ if $\pi_\theta(a_t|s_t) = 1 \forall t \leq |a|$. By definition $S_{PL} : \pi_\theta(a_t|s_t) = 1 \forall t \leq |a| \forall a \in \mathcal{A}$, thus $\nabla_\theta L_{PL}(\theta, q) = 0, \nabla_\theta L_{NL}(\theta, q) = 0$ at $S_{PL}$, hence $\nabla_\theta L_{CL}(\theta, q) = 0$.

Similarly, we know that $S_{\text{NL}} : \pi_\theta(a_t|s_t) = \frac{1}{|V|} \forall t \leq |a|, \forall a \in \mathcal{A}$. As shown in Eq. 30 this implies $\nabla_\theta L_{NL}(\theta, q) = 0$. Further since $\pi_\theta(a_t|s_t) = \frac{1}{|V|} \forall t \leq |a|; \implies \frac{|A^+|}{|A|} \approx 0$. Thus at $S_{NL}$ :

$$\nabla_\theta L_{CL}(q) = \frac{|\mathcal{A}^+|\nabla_\theta L_{\text{PL}}(q) + |\mathcal{A}^-|\nabla_\theta L_{\text{NL}}(q)}{|\mathcal{A}|} \tag{40}$$

$$\nabla_\theta L_{CL}(q) = \frac{|\mathcal{A}^+|\nabla_\theta L_{\text{PL}}(q)}{|\mathcal{A}|} \approx 0 \tag{41}$$

This shows that $S_{NL}$ and $S_{PL}$ are critical solutions for $L_{CL}$. $\qquad\square$

## L  DETAILS ON DATASETS AND TRAINING

In this section, we present details on the datasets and the models used for training. We train Qwen2.5-7B base, Qwen2.5-7B Instruct, and Llama3.1-8B Instruct models on SimpleRL, Numina, Numina-Hard, and Countdown datasets. All the experiments are done using a batch size of 128, with a constant learning schedule and a learning rate of 1e-6. We use Adam optimizer with no weight decay. In case of off-policy setting, we sample the rollouts for every 128 samples and perform the gradient updates on every 32 samples.

SimpleRL consists of 8024 samples taken from GSM8K and Math datasets, Numina consists of approximately 83k problems, where as Numina-Hard consists of around 12k problems which are sampled from Numina by ensuring that Qwen2.5-7B base fails on them in both attempts made from it. Countdown consists of around 5k problems. For Xiong et al. (2025); Prabhudesai et al. (2025) and Zhu et al. (2025) we verify our analysis using their version of code as well.

