# OpenReview forum: "Towards demystifying the optimization landscape of RLVR methods"
_ICLR.cc/2026/Conference — Submitted to ICLR 2026_

### Official Review · Reviewer_HVFW · 2025-10-29

**Soundness:** 3
**Presentation:** 3
**Contribution:** 2
**Rating:** 4
**Confidence:** 3

**Summary:**

This paper demystifies the GRPO algorithm, widely used for LLM reasoning, by re-framing its on-policy optimization as a weighted combination of maximizing the likelihood of correct rollouts and minimizing that of incorrect ones. The authors demonstrate that optimizing either of these objectives independently leads to unstable bad minima, characterized by either entropy collapse or entropy explosion. The study reveals that GRPO's success stems from its specific advantage calculation and the use of clipping, which work together to stabilize the training process and prevent convergence to these undesirable solutions.

**Strengths:**

1. The paper is well-written and easy to follow.

2. The paper offers an intuitive way to understand the optimization of RLVR methods. It reframes the objective as a balance between two competing forces: maximizing the likelihood of correct responses and minimizing the likelihood of incorrect ones. This model clearly explains why simpler methods are unstable, showing they can lead to "length collapse" or "length explosion" when one force overpowers the other.

3.  The paper presents a practical discovery: off-policy training, when combined with clipping, can be significantly more stable than its on-policy counterpart. This finding challenges common assumptions about on-policy methods and provides a valuable, counter-intuitive insight for practitioners.

**Weaknesses:**

1. While the paper's analysis of the problem is insightful, the solution it proposes—using token-level normalization—is not new. As the authors acknowledge, this technique is already a key component in several other recent and successful methods. Therefore, the paper's contribution feels more like a strong explanation for why an existing method works, rather than a new solution derived from its analysis indicating that, while the analysis is valuable, it doesn’t lead to any substantively new insight or proposal derived from the authors’ interpretation and findings.

2. The paper's core claims are about training instability and collapse, which are phenomena often highly sensitive to random seeds and initialization. The authors state that all experiments were run only once. This could be regarded as a significant limitation considering the instability of GRPO algorithm. The claims would be much stronger if they were supported by results averaged over multiple runs(seeds) to show variance and confirm that the observed collapses are consistent.

3. While the paper identifies that clipping is the key to off-policy stability, it admits a "do not have a complete understanding"  of the underlying mechanism why it works. Understanding how clipping "induces stability" is left as an "interesting future direction", making the paper's "demystification" partially incomplete.

4. I believe that "RLVR method" term in the title is too broad considering the algorithm handled in the paper. I would recommend that the authors consider changing the title. (for instance, changing "RLVR method" into GRPO)

**Questions:**

1. The paper attributes the observed instability primarily to the 'on-policy setting' itself. However, the GSPO[1] posit that the instability in GRPO does not stem from the on-policy setting, but rather from the high-variance noise introduced by its fundamentally flawed 'token-level importance sampling' design. GSPO algorithm is also an off-policy method, yet it achieves stable training where GRPO fails. Do the authors believe their perspective—that clipping is the critical component for stability in combined with off-policy methods—generalizes to sequence-level algorithms like GSPO as well? In other words, under a sequence-level optimization framework, do you still consider the clipping mechanism to be equally critical for maintaining stability than on-policy setting??

[1] Chujie Zheng, Shixuan Liu, Mingze Li, Xiong-Hui Chen, Bowen Yu, Chang Gao, Kai Dang, Yuqiong Liu, Rui Men, An Yang, Jingren Zhou, & Junyang Lin. (2025). Group Sequence Policy Optimization.

---

> ### Author Response · Authors · 2025-11-26
> **Rebuttal**
>
> We thank the reviewer for their detailed comments. We request the reviewer to kindly check the general response and our revised version, where we have highlighted the changes in blue. We answer specific questions pointed out by the reviewer below:
>
> * **Novelty in token level normalization:** We would like to clarify that we did not intend to claim token level normalization as a newly proposed approach, but instead aim to understand how it induces stability in the training process. Acknowledging this, we have improved the presentation in Sec. 6 of the main paper. We highlight that using token level normalization changes the nature of critical solutions in the loss landscape by introducing a conflict between the properties of the critical solutions and the length of the rollouts. That is high entropy critical solutions achieve low loss when the output length is small, whereas low entropy solutions achieve low loss when the output length is large. This conflict makes it difficult for the model to converge to the bad critical solutions. *To demonstrate this clearly, we have added Fig.4 and Fig.7 in the main paper, where we can see the drastic difference between the rollouts at the time of collapse when using (Fig. 7) vs not using (Fig. 4) token level normalization*. We highlight that this insight provides a new way to understand the mechanism which helps token level normalization induce stability.
> * **Reruns to confirm variance in runs:** We agree that reruns are helpful in studies like this. However, we would like to mention that each of the runs presented in this work take more than 96 hours to do on 4 A100 GPUs. Due to limited resources, we were not able to do reruns. However, we would like to clarify that we haven't cherry picked any of the results presented in this work, which makes us believe that the trend presented in this work won't vary significantly on doing reruns. We will add additional results with reruns in the revised version of our work.
> * **Lack of understanding on how clipping induces training stability:** We agree that the current version doesn't clarify the mechanism using which clipping helps in inducing training stability and for the same reason we highlight this as an interesting future direction.
> * **Suggestion on changing paper's title:** We indeed agree with reviewer's suggestion and we have modified the title of the paper to *Towards understanding the optimization landscape of GRPO and its variants*.
> * **GSPO being more stable than GRPO:** GSPO formalizes GRPO into a contextual bandits setting and considers the entire sampled rollout as an action, whereas GRPO considers each token's prediction as an action. GRPO makes further assumptions:
>     * **Advantages are constant across the states:** GRPO assumes that the advantages are same across different states in the rollout.
>     * **Q-value function is approximated by a single rollout:** GRPO approximates the Q-value function using a single rollout sampled at the starting state of the MDP.
>
>     Due to these assumption, GRPO loses any guarantee of value maximization in the MDP. However, since GSPO doesn't need to make these approximations (as it considers the contextual bandit setting), it gives a good estimate of the true value function of the action. We believe that this is the primary reason for its enhanced stability. In case of simplified versions of GRPO (Zhu et al., 2025; Xiong et al., 2025), we demonstrate that clipping primarily contributes in stabilizing the training. However, if the on-policy version of the algorithm is already stable then the role of clipping becomes unclear. Therefore, clipping is not expected to be equally important for GSPO as compared to GRPO.
>
>
> We thank the reviewer for their detailed comments which have helped in imporving the clarity of our work. We look forward to further discussions.
>
>  **References:**
>
> * Wei Xiong, Jiarui Yao, Yuhui Xu, Bo Pang, Lei Wang, Doyen Sahoo, Junnan Li, Nan Jiang,
>   Tong Zhang, Caiming Xiong, and Hanze Dong.
>   A minimalist approach to LLM reasoning: from rejection sampling to REINFORCE. 2025.  URL: https://arxiv.org/abs/2504.11343
>
> * Xinyu Zhu, Mengzhou Xia, Zhepei Wei, Wei-Lin Chen, Danqi Chen, and Yu Meng.
>   The surprising effectiveness of negative reinforcement in LLM reasoning: arXiv preprint arXiv:2506.01347, 2025.

---

### Official Review · Reviewer_f2dX · 2025-10-30

**Soundness:** 2
**Presentation:** 2
**Contribution:** 3
**Rating:** 4
**Confidence:** 3

**Summary:**

The paper investigates the GRPO algorithm for optimizing LLMs in a reinforcement learning with verifiable rewards (RLVR) setting. The findings highlight differences between on-policy and off-policy training, the importance of training both on positive and negative samples, and the importance of likelihood ratio clipping. The authors show that removing certain components can lead to training instabilities.

**Strengths:**

The paper provides an in-depth analysis of different components of the RLVR optimization of LLMs. Evaluation is done on different datasets, and the trends seem to be similar across the different datasets.

**Weaknesses:**

Certain parts of the paper are not very clear (see Questions).

Furthermore, the paper claims that "PPO collapses in on-policy setting". This claim seems misleading. On-policy PPO here means that $\pi_{\theta_\mathrm{old}} = \pi_\theta$, which means that the ratio $\pi_\theta(a|s) / \pi_{\theta_\mathrm{old}}(a|s)$ is always 1. This, in turn, means that the clipping is never active, and what is left is essentially a vanilla policy gradient algorithm (similar to on-policy GRPO in eq. 6). Since the clipped loss is the central component of PPO, this notion of "on-policy PPO", therefore, does not bear a lot of resemblance to PPO anymore.

**Questions:**

1. Figure 1 is not clear. What exactly is 1(a) showing? Is this just an illustration, or is this some visualization of a loss landscape? Which loss is shown here? I assume the positive + negative likelihood loss? What is C_DL? The caption only says that it "leads to improved performance". (b) and (c) are also not clear. What do the black dots, arrows, and orange / brown curves represent? What are the little lines on the top right of the curves?

2. The caption of Figure 1 states that "importance sampling reduces the norm of the gradients, resulting in slower convergence". However, I did not find any data in the paper backing this up.

3. C_NL and C_PL are the minima of L_NL and L_PL, respectively. This should not directly mean that the they are also minima of L_CL, which is the (weighted) sum of L_PL and L_NL, but the paper often treats these points as minima of L_CL or even GRPO's loss (e.g., in section 4.3 "Clearly, the two critical solutions C_PL and C_NL [...], are critical solutions of L_CL as well" or in the Figure 1(a)). I would appreciate it if the authors could clarify why these points are also minima of L_CL / GRPO's loss.

4. Section 4.3.: At some point, the critical solutions are referred to as S_PL and S_NL, instead of C_PL and C_NL. Do both refer to the same thing?

5. The paper repeatedly claims that it is surprising that off-policy training is more stable than on-policy training, e.g., in section 7. To me, this does not seem surprising since in on-policy training, problematic updates have a very immediate and potentially catastrophic effect on the training data for the next updates, which makes it hard to recover from the suboptimal update. In the off-policy case, the data-collecting policy and the optimized policy are somewhat decoupled, which can help with this problem. Increasing the stability of training was also the reason why, e.g., DQN uses a replay buffer. I would appreciate it if the authors could elaborate on why they expect on-policy training to be more stable than off-policy training.

Typos:

1. Section 4.1: "decease" --> "decrease"

2. Section 4.1: "in length of model's entropy"

3. Section 4.3.: "the gradients becomes zero" --> "the gradients become zero"

---

> ### Author Response · Authors · 2025-11-26
> **Rebuttal**
>
> We thank the reviewer for their detailed comments. We request the reviewer to kindly check the general response and our revised version, where we have highlighted the changes in blue. We answer specific questions pointed out by the reviewer below:
> * **On-Policy PPO is misleading:** We agree with reviewer's comment that analyzing PPO in on-policy setting removes the key essence of being close to the old policy, which ensures that the updates remain small, thereby inducing stability. For the same reason, we have moved Sec. 6 to the appendix and removed the discussion about PPO from the main paper.
> * **Concerns about Fig. 1:** We agree that Fig. 1 in the previous version was confusing. Fig. 1 (a) is an illustrative diagram and doesn't reflect the true loss landscape of a model. The y axis corresponds to the loss function of GRPO, which has critical solutions corresponding to maximum entropy ($C_{NL}$), minimal entropy ($C_{PL}$) and desired entropy and likelihood ($C_{DL}$). Here, $C_{NL}$, and $C_{PL}$ corresponds to the two collapses and $C_{DL}$ corresponds to an illustrative solution that GRPO converges to. In the b part, the black dots are used to represent a neural network at some time stamp during training. The length of the arrows represent the norm of the gradients in the direction of either of the critical solutions given by $C_{PL}$ and $C_{NL}$. A longer arrow represents larger norm of the gradient. A curve represents the 2-D projection of a critical solution projected along the direction of maximum curvature. We have removed the c part of the figure for clarity purposes.
> * **Evidence on importance sampling reduces gradient norms:** We indeed observe this in our experiments, where the average norm of the gradients decreases on using off-policy training with importance sampling as compared to removing the importance sampling term from GRPO's objective function. We will add additional results related to this in the revised version of our draft.
> * **$C_{NL}$ and $C_{PL}$ as critical solutions to $L_{CL}$ and GRPO:** We agree that it might not be trivial to argue that $C_{NL}$ and $C_{PL}$ will be the critical solutions of $C_{CL}$. We have added a proof showing this in the appendix Sec. K (line 1330 onwards). We would encourage the reviewer to have a look at this section.
> * **Distinction between $S_{PL}$, $S_{NL}$, and $C_{PL}$, $C_{NL}$:** $S_{NL}, C_{NL}$, and $S_{PL}, C_{PL}$ do not refer to the same solutions. $C_{PL}, C_{NL}$ correspond to the empirically realizable versions of $S_{PL}$ and $S_{NL}$. This means that $S_{NL}$ and $S_{PL}$ are the critical solutions expected to be learned when the number of rollouts are infinite, but since we sample only a few rollouts in practice, the critical solutions learned need not be exactly same. This is precisely the reason why we use $C_{PL}, C_{NL}$ as empirically realizable versions of $S_{PL}, S_{NL}$ respectively.
> * **Enhanced stability of on-policy as compared to off-policy training:** We would like to thank the reviewer for pointing this out, and we request the reviewer to check our common reply (point: 1), where we have addressed this point in detail.
>
> We thank the reviewer for their detailed comments which have helped in imporving the clarity of our work. We look forward to further discussions.

---

### Official Review · Reviewer_LhWV · 2025-11-01

**Soundness:** 3
**Presentation:** 2
**Contribution:** 1
**Rating:** 2
**Confidence:** 5

**Summary:**

This paper starts by analyzing GRPO. The authors "demystify" it by breaking its objective function into two parts: a positive term that maximizes the likelihood of correct answers and a negative term that minimizesthe likelihood of incorrect answers.

The paper's key findings are:
* GRPO is stable because its advantage calculation acts as a built-in stabilizer, preventing it from converging to these bad solutions.
* PPO's value estimators have "large errors from the true estimates".
* Off-policy training with clipping is more stable than standard on-policy training, as clipping also prevents these collapses.

**Strengths:**

1. The paper is easy to read and easy to follow.
2. The hypotheses raised by the authors are sound and accompanied by experiment observations.
3. The authors carry out experiments on multiple datasets, which cross validate their ideas.

**Weaknesses:**

1. The instabilities and collapses the authors identify are well-known failure modes that arise from the combination of function approximation, bootstrapping and off-policy learning, namely deadly triad [1].
2. PPO's value estimators having "large errors from the true estimates" is also a well known issue as critic models tend to overestimate values [2].
3. Becuase of 1 and 2, I question the novelty of this paper. i.e., I don't think this paper has enough new insights, nor does this paper offers novel solutions (clipping is not novel) to the above findings that achieve SOTA results.
4. I also think Figure 1 is really confusing. Why on the loss surface, C_DL is the most optimal trajectory but C_NL leads to the minimum entropy? Also I don't understand the illustration of Figure 1(c) completely.

[1] Van Hasselt H, Doron Y, Strub F, Hessel M, Sonnerat N, Modayil J. Deep reinforcement learning and the deadly triad. arXiv preprint arXiv:1812.02648. 2018 Dec 6.

[2] Van Hasselt, H., Guez, A., & Silver, D. (2016, March). Deep reinforcement learning with double q-learning. In Proceedings of the AAAI conference on artificial intelligence (Vol. 30, No. 1).

**Questions:**

See the above weaknesses.

---

> ### Author Response · Authors · 2025-11-26
> **Rebuttal**
>
> We thank the reviewer for their comments. We request the reviewer to kindly check the general response and our revised version, where we have highlighted the changes in blue. We answer specific questions pointed out by the reviewer below:
> * **Novelty on observed collapses and relationship with deadly triad:** We agree that deadly triad is studied well enough in the RL literature, and for the same reason we indeed observe lower stability when training in off-policy setting (See Fig. 9 in the updated version) as compared to on-policy setting (Fig. 2). But in this work, we highlight that instabilities can occur in on-policy settings of several simplified versions of GRPO, where due to on-policy, these settings do not form a deadly triad. However, we find that GRPO always remains stable in the on-policy setting. We believe this finding is interesting as all other methods (including PPO) are unstable in the on-policy setting. To understand the cause of this, we further dive into understanding the reweighting mechanism used by GRPO and how it enforces training stability.
> * **Novelty on PPO's value estimators having high errors:** We indeed agree with the reviewer that our observation regarding PPO's value estimators having high error is not new. Indeed Kazemnejad et al. (2024) highlights this as well for LLMs. We would like to clarify that we are not claiming this observation to be novel, and the primary reason we mention this is to explain the collapses we observe with PPO. We would like to request the reviewer to check the common reply where we clarify the novelty of our work.
> * **Clarity on Fig. 1:** We have modified Fig. 1 and improved its clarity. We would like to clarify that this figure is drawn merely for illustrative purposes and doesn't reflect the true loss landscape of the model. We agree that the c part of the figure was confusing, and we have removed it in our revised version.
>
> We thank the reviewer for their detailed comments which have helped in imporving the clarity of our work. We look forward to further discussions.
>
>
>
> **References:**
>
>
> * Amirhossein Kazemnejad, Milad Aghajohari, Eva Portelance, Alessandro Sordoni, Siva Reddy,
>   Aaron Courville, and Nicolas Le Roux.
>   Vineppo: Refining credit assignment in RL training of LLMs: arXiv preprint arXiv:2410.01679, 2024.

---

### Official Review · Reviewer_meCw · 2025-11-03

**Soundness:** 2
**Presentation:** 1
**Contribution:** 1
**Rating:** 2
**Confidence:** 4

**Summary:**

This paper attempts to deliver concrete insights into the specific reasons that GRPO has proven to be an effective RLVR optimization technique for reasoning. The work presents a series of ablative experiments, focused primarily on elements of policy gradient algorithms deriving from PPO (e.g. clipping, reweighting with advantage functions, etc) to demonstrate the differences in training stability as well as investigate the differences between on- and off-policy training.

**Strengths:**

I am grateful for works such as this paper that attempt to explore the specific contributions of known methods. These papers are critically important for improved scientific understanding and help to develop improved algorithmic approaches. This papers sets an ambitious goal to study the effects of various elements of GRPO across datasets, model family as well as training paradigm.

I felt that the most interesting section of the analysis came in Section 4.3 where the paper dug into the reweighting mechanism of GRPO which appears to balance between negative and positive generations. This led into a potentially deeper insight into the approximate advantage function used in GRPO. I was left wishing that the majority of the paper focused on this analysis rather than trying to cover every aspect of what the authors identified as contributions of GRPO for RLVR.

**Weaknesses:**

Overall, I'm not entirely sure that this paper introduced any novel insights beyond those identified in the literature they cited that have deeply investigated specific aspects of GRPO. While these unified surveys can be really useful if done rigorously and thoroughly, I do not feel that this paper meets those criteria. In many ways, it feels that the paper is trying to do too much at once and comes across as unfocused. This led to an overly distracted presentation in the paper where proposed insights are not deeply motivated or justified.

Perhaps the major flaw of this work is that it comes across as unaware of the RL theory underlying policy gradient methods, where many of the proposed insights about stability, variance and the trade-offs between on- and off-policy training have been well understood for decades. The discussion in the final paragraph largely restates the principled motivations that led to the development of trust region policy gradient approaches (of which TRPO and, later, PPO derive from). Policy gradient methods have been known to be reweighted MLE objectives since their introduction. Reformulations of policy gradient methods as weighted regression have further established this relationship (see, Peters and Schaal, "Using Reward-weighted Regression for Reinforcement Learning of Task Space Control" (2007)).

Within the perspectives of LLM Reasoning, where I feel the authors are largely situated within, I think that there are some errors in the proposed development of the component losses for positive and negative generations individually. This is especially true within the lens that these are ablations of the GRPO objective since there is no controlling of simplified advantage function being a relative estimate of the group. Without this, it's not clear whether or not $L_{PL}$ and $L_{NL}$ are valid comparisons. There was not sufficient justification or grounding of these re-derivations to ensure that the policy gradient objective was not affected through the use of a biased baseline (for more about the use of baselines in policy gradient approaches, I'd highly recommend this recent blog post: https://fatemi.github.io/posts/pg-baseline/).

The GRPO objective provided in Equation 3 is incomplete as there is no aggregation over the group, including length normalization. Please revisit Shao, et al (2024). This omission leads me to have less confidence in the remaining development of the various objectives and the resulting analyses. This concern extends to the imprecise manner in which the advantage approximations are made throughout Section 4.

The insights from Takeaway 2 are well established in the community, particularly those surrounding the effect of negative gradients. Please see the following two papers for detailed analysis.
- Setlur and Yang, et al (2025), "e3: Learning to Explore Enables Extrapolation of Test-Time Compute for LLMs"
- Fatemi, et al (2025), "Concise Reasoning via Reinforcement Learning"

**Questions:**

I do not have any further questions for the authors beyond the concerns raised in the "Weaknesses" section above.

---

> ### Author Response · Authors · 2025-11-26
> **Rebuttals (1/2)**
>
> We thank the reviewer for their detailed comments. We request the reviewer to kindly check the general response and our revised version, where we have highlighted the changes in blue. We answer specific questions pointed out by the reviewer below:
> * **Lack of focus and novelty:** We thank the reviewer for highlighting this, and we indeed agree that writing needed to demonstrate more focus. In this regard, we have made the following changes in the revised version:
>     * Highlighted the motivation for analyzing clipping and on-policy setting in Sec. 1, and Sec. 2.1 of the main paper.
>     * Improved the clarity in Fig. 1.
>     * Added more discussion about comparison between GRPO and different policy gradient methods (Reinforce in particular) in Sec. 2.1.
>     * Added more discussion about motivation for analyzing $L_{PL}$ and $L_{NL}$ in Sec. 4.
>     * Moved Sec. 6 to appendix, and toned down our claims about stability induced by clipping in off-policy setting.
>     * Added Fig.4 and Fig.7 to analysing the rollouts at the time of collapse, when using vs not using token level normalization.
>
>     Regarding novelty, we request the reviewer to kindly check our common reply, where we have explained the novelty of work in detail.
> * **Lack of motivation behind studying comparison between stability of on and off policy methods for language models:** We would like to thank the reviewer for pointing this out, and we request the reviewer to check our common reply (point: 1), where we have addressed this point in detail.
> * **Policy gradient as reweighted version of MLE:** We request the reviewer to check our common reply (point: 2), where we have addressed this point in detail.
> * **Motivation behind $L_{PL}$ and $L_{NL}$:** We agree that the previous version did not highlight enough motivation behind studying $L_{PL}$ and $L_{NL}$. To improve the clarity, we have added additional discussion motivating the exploration of $L_{PL}$ and $L_{NL}$ in Sec. 4.
> > The key motivation behind formulating $L_{PL}$ and $L_{NL}$ is based on the observation in Sec. 3 showing that GRPO can be interpreted as reweighted version of likelihood maximization and minimization on rollouts with correct and incorrect answers.
>
> Thus, in order to understand GRPO's working, it becomes important to understand each of these components (i.e. $L_{PL}$ and $L_{NL}$) separately.  We also highlight that a few recent works (Zhu et al., 2025; Xiong et al., 2025) trying to simplify GRPO's objective function, optimize $L_{NL}$, and $L_{PL}$ separately.
> These methods demonstrate stable increase in performance of the policy, but understanding the underlying reasons for the improvement in performance remains unclear. These two points were our the main sources of motivation for analyzing $L_{PL}$ and $L_{NL}$. We find that using clipping with the objective functions proposed in (Zhu et al., 2025; Xiong et al., 2025) is the hidden gem which makes these methods stable.
>
> * **Use of unbiased baselines in $L_{PL}$ and $L_{NL}$:** In regards to the blogpost (https://fatemi.github.io/posts/pg-baseline), we agree that a key motivation to define advantages instead of maximizing the Q-values when using policy gradients was to reduce the variance of gradients and make training stable. Therefore, another way to look at the objective functions $L_{PL}$ and $L_{NL}$ can be to consider that we are using a zero baseline instead of the true value function, where the Q-value function is approximated by a single rollouts (as also assumed in GRPO), where in case of $L_{PL}$ the reward is $0$ and $1$ and in case of $L_{NL}$ the reward is defined as $0$ and $-1$ for correct and incorrect answers respectively. Since the zero baseline doesn't depend on the sampled action, $L_{PL}$ and $L_{PL}$ use unbiased baselines.
>
> * **Clarification on Eq. 3:** We agree that we did not define the advantages properly in the previous version, which led to the confusion. Please check the definition of advantages (line 160) in our revised version, which makes Eq. 3 justifiable. Indeed, we have also written an empirical expectation version of Eq.3 as Eq.7 which demonstrates the group aggregation operation for a single prompt. We hope this clarification helps the reviewer in developing a better understanding about development of the various objectives introduced in our paper.

---

> ### Author Response · Authors · 2025-11-26
> **Rebuttals (2/2)**
>
> * **Novelty on Takeaway 2:** We agree with the reviewer, that it is well known that negative gradients help in improving the exploration and hence the entropy of the model, whereas the positive gradients help in sharpening, thereby decreasing model's entropy. However, in this work we demonstrate an extreme case of them, which leads to model collapse. In addition, we characterize the form as well as stability of the critical solution corresponding to collapses. As shown in lines 313-348, we also highlight the dynamics of the rollouts at the time of collapse, where on maximizing the likelihood for correct samples lead to answers without any reasoning, whereas minimizing the likehood for incorrect samples makes intiially makes the model output random tokens with high entropy. However, this critical solution is unstable, as a result the model converges into outputting repeated tokens where it's entropy decreases. We believe that this insight is new within the scope of community working on LLM reasoning. *To demonstrate this more clearly, we have added Fig. 4 in the main paper.*
>
> We thank the reviewer for their detailed comments which have helped in imporving the clarity of our work. We look forward to further discussions.
>
>  **References:**
>
> * Zhihong Shao, Peiyi Wang, Qihao Zhu, Runxin Xu, Junxiao Song, Xiao Bi, Haowei Zhang,
>   Mingchuan Zhang, Y. K. Li, Y. Wu, and Daya Guo.
>   DeepSeekMath: Pushing the limits of mathematical reasoning in open language models. 2024.  URL: https://arxiv.org/abs/2402.03300
>
> * Wei Xiong, Jiarui Yao, Yuhui Xu, Bo Pang, Lei Wang, Doyen Sahoo, Junnan Li, Nan Jiang,
>   Tong Zhang, Caiming Xiong, and Hanze Dong.
>   A minimalist approach to LLM reasoning: from rejection sampling to REINFORCE. 2025. URL: https://arxiv.org/abs/2504.11343
>
> * Xinyu Zhu, Mengzhou Xia, Zhepei Wei, Wei-Lin Chen, Danqi Chen, and Yu Meng.
>   The surprising effectiveness of negative reinforcement in LLM reasoning: arXiv preprint arXiv:2506.01347, 2025.

---

### Author Response · Authors · 2025-11-26
**General Comments (1/2)**

We thank the reviewers for their detailed comments, which will certainly help us in improving the quality of our work. We are glad that the reviewer meCw found the direction of our work interesting, along with the highlighted reweighted mechanism in GRPO (meCw, HVFW). Additionally, reviewers LhWV and HVFW found the paper well written and easy to follow, with evaluations across multiple datasets and models (LhWV, HVFW, f2dX), providing an in-depth understanding on different components of RLVR methods (HVFW, f2dX).

We have incorporated their suggestions and made the following changes in the revised version. We encourage the reviewers to have a look at the revised version where we have highlighted the changes in blue:

1) **Enhanced stability of Off-Policy on using Clipping:**  We agree with reviewer meCw and f2dX that our claim about finding off-policy methods with clipping being more stable than their corresponding on-policy version was overstated, and we have toned it down by removing Sec. 5 and adding more discussion in Sec. 4.2 instead. We have also highlighted the motivation for analyzing the role of clipping in off-policy methods in Sec. 2.1, which we also highlight below. We agree that we should include more discussion about PPO and TRPO, which we have now indeed done in Sec. 1 and Sec. 2.1. We indeed agree that the key motivation behind TRPO, and later PPO was to enhance the stability of deep RL methods including the **on-policy** ones, where TRPO proposed a policy improvement guarantee in every iteration given that the policy remains in local neighborhood of the old policy. PPO realizes this constraint by proposing clipping and KL divergence between the new and the old policy as a heuristic. GRPO also adopts this heuristic, and so do other simplified versions of GRPO (Zhu et al., 2025; Xiong et al., 2025).

>However, the incorporation of this motivation for different RLVR algorithms used for fine-tuning language models still remains questionable as these methods are significantly different from PPO.

This is because PPO aims to maximize a lower bound on the value functions for the states in the MDP, while ensuring that the policy being optimized is close to a base policy. This lower bound maximization is what leads to policy improvement guarantee.
> But minimizing or maximizing the likelihood only on the incorrect (Zhu et al., 2025), or correct (Xiong et al., 2025) rollouts respectively (or both) as done by GRPO and its simplified versions, shouldn't necessary lead to maximization of the value function of the states in the MDP in every iteration as here we are not maximizing a lower bound on the value functions of the states (as done in PPO).

This is the central argument which makes the applicability of the policy improvement guarantee given by TRPO in each iteration for GRPO and its simplified versions questionable. As a result, we believe that there is insufficient motivation behind using the clipping objective in GRPO and its simplified variants. To understand this in detail, we analyze if clipping can really induce stability over the on-policy versions of these methods. As highlighted in Sec. 4.2, we find that while GRPO is already stable in on-policy setting, in case of its simplified variants (Zhu et al., 2025; Xiong et al., 2025), clipping indeed helps in inducing enhanced stability. We agree with the reviewer, that our work missed highlighting this key motivation for studying on-policy methods which potentially led to some confusion. But we have added discussions about it in the revised version (Sec. 2.1, 4.2).

2) **GRPO as reweighted version of MLE:** We agree with the reviewer meCw that policy gradient methods can be seen as a reweighted version of MLE, and we have strengthened this point in the revised version by adding a small discussion in Sec. 2.1, which we highlight below. It is very much evident that algorithms like Reinforce can be seen as maximizing the likelihood with reweighting at token level. However, in this work, we argue that the setting is much more simplified when using GRPO in language models. As highlighted in Sec. 2, there are two key assumptions that are baked in GRPO's objective function:
>* Advantages are constant across the states: GRPO assumes that the advantages are same across different states in the rollout.
 >* Q-value function is approximated by a single rollout: GRPO approximates the Q-value function using a single rollout sampled at the starting state of the MDP.

  Note that the two assumptions highlighted above makes GRPO a sample level reweighted MLE estimator instead of the token level as in the case of Reinforce. Thus, GRPO's optimization is far from maximizing the true value function of the MDP's states which is the ideal goal of any policy gradient method.

---

> ### Author Response · Authors · 2025-11-26
> **General Comments (2/2)**
>
> 3) **Novelty in token level normalization:** We would like to clarify that we did not intend to claim token level normalization as a newly proposed approach, but instead aim to understand how it induces stability in the training process. Acknowledging this, we have improved the presentation in Sec. 6 of the main paper. We highlight that using token level normalization changes the nature of critical solutions in the loss landscape by introducing a conflict between the properties of the critical solutions and the length of the rollouts. That is high entropy critical solutions achieve low loss when the output length is small, whereas low entropy solutions achieve low loss when the output length is large. This conflict makes it difficult for the model to converge to the bad critical solutions. *To demonstrate this clearly, we have added Fig.4 and Fig.7 in the main paper, where we can see the drastic difference between the rollouts at the time of collapse when using (Fig. 7) vs not using (Fig. 4) token level normalization.* We highlight that this insight provides a new way to understand the mechanism which helps token level normalization induce stability.
>
> 4) **Clarity on Fig.1:** We agree with the reviewers that Fig.1 was unclear in terms of the presentation and we have modified it in the revised version. We have also removed the part c of the figure for clarity purposes.
>
> We hope that the changes highlighted above will help in imporving the clarity of our work.
>
>
>
>  **References:**
>
> * Zhihong Shao, Peiyi Wang, Qihao Zhu, Runxin Xu, Junxiao Song, Xiao Bi, Haowei Zhang,
>   Mingchuan Zhang, Y. K. Li, Y. Wu, and Daya Guo.
>   DeepSeekMath: Pushing the limits of mathematical reasoning in open language model,  2024. URL: https://arxiv.org/abs/2402.03300
>
> * Wei Xiong, Jiarui Yao, Yuhui Xu, Bo Pang, Lei Wang, Doyen Sahoo, Junnan Li, Nan Jiang,
>   Tong Zhang, Caiming Xiong, and Hanze Dong.
>   A minimalist approach to LLM reasoning: from rejection sampling to REINFORCE, 2025.  URL: https://arxiv.org/abs/2504.11343
>
> * Xinyu Zhu, Mengzhou Xia, Zhepei Wei, Wei-Lin Chen, Danqi Chen, and Yu Meng.
>   The surprising effectiveness of negative reinforcement in LLM reasoning: arXiv preprint arXiv:2506.01347, 2025.

---

### Author Response · Authors · 2025-11-26
**Clarification on Novelty**

We would like to highlight the novelty and key contributions of this work:

1) **Demonstration of the assumptions baked in GRPO's objective function:** In Sec. 2, we highlight two key assumptions that are baked in GRPO's objective function without explicitly acknowledging them in Shao et al. (2024). This simplifies GRPO's objective function in on-policy setting, into a reweighted version of iterated MLE.

2) **Role of the reweighting mechanism to stabilize GRPO in on-policy setting:** In Sec. 4.3, we highlight that GRPO is indeed very stable in on-policy setting, despite its simplified versions (Zhu et al., 2025; Xiong et al., 2025) being unstable. This stability is induced by the reweighting mechanism in GRPO's advantages, where, as the model converges to either of the bad critical solutions, its gradients in that direction becomes small, while increasing in the direction of the other critical solution. This self stabilizing mechanism in GRPO makes it stable even in the on-policy setting.

3) **Demonstrating the utility of clipping in stabilizing training of objectives significantly different from PPO:** Although clipping was proposed by PPO to induce training stability, we highlight that GRPO, and its simplified versions do not maximize a strict lower bound on the value function of the states in the MDP as there are several simplifications and approximations considered. Thus the motivation of using clipping with their objective functions is not well grounded. Here, we demonstrate that clipping indeed helps in improving the stability of simplified versions of GRPO: (Zhu et al., 2025; Xiong et al., 2025).

4) **Characterizing how token level normalization helps induce stability:** In Sec. 6, we provide a new lens on understanding how the existing token normalization methods help induce training stability. We highlight that using token level normalization changes the form of critical solutions in the loss landscape by introducing a conflict, which makes it difficult for the model to converge to them. This induces an inherent training stability in on-policy setting, which we verify empirically in Fig. 6 and 7.


**References:**

* Zhihong Shao, Peiyi Wang, Qihao Zhu, Runxin Xu, Junxiao Song, Xiao Bi, Haowei Zhang,
  Mingchuan Zhang, Y. K. Li, Y. Wu, and Daya Guo.
  DeepSeekMath: Pushing the limits of mathematical reasoning in open language models, 2024.  URL: https://arxiv.org/abs/2402.03300

* Wei Xiong, Jiarui Yao, Yuhui Xu, Bo Pang, Lei Wang, Doyen Sahoo, Junnan Li, Nan Jiang,
  Tong Zhang, Caiming Xiong, and Hanze Dong.
  A minimalist approach to LLM reasoning: from rejection sampling to REINFORCE, 2025.  URL: https://arxiv.org/abs/2504.11343

* Xinyu Zhu, Mengzhou Xia, Zhepei Wei, Wei-Lin Chen, Danqi Chen, and Yu Meng.
  The surprising effectiveness of negative reinforcement in LLM reasoning, arXiv preprint arXiv:2506.01347, 2025.

---

### Meta-Review · Area_Chair_Q52J · 2026-01-06

**Summary:**

This paper analyzes why GRPO is successful for RLVR optimization in LLM reasoning. The authors analyze why elements in GRPO could improve training stability.

The reviewers have several concerns in their initial reviews, including lack of novelty (in particular, some insights/takeaway messages of the current paper is by now well-known in the community or in classical RL theory) and overstating the effect of off-policy learning with clipping.

**Reviewer Concerns:**

The authors tried to address the concern of overstating the effect of off-policy learning with clipping by toning that part down and removing Sec. 5. The authors further argue that although some insights/takeaway messages of the current paper is by now well-known in the community or in classical RL theory, those analysis apply mostly to algorithms like REINFORCE and PPO, while the current paper focuses on GRPO which is quite different / much more simple.

Well these modifications / arguments might have addressed part of the reviews' concern, I believe the changes are significant enough so that another round of peer-review is necessary.

**Reviewer Scores:**

Reviewer LhWV and Reviewer might have changed their scores from 2 to 4 if they were able to participate fully in the discussion. However, it is hard to imagine that the majority of the reviewers would increase their scores to positive ones even with full discussion.

---

### Decision · Program_Chairs · 2026-01-26

Reject